# Uniqueness and Complexity of Inverse MDP Models

## Abstract

What action sequence $aa'a''$ was likely responsible for reaching state $s'''$ (from state $s$) in 3 steps? Addressing such questions is important in causal reasoning and in reinforcement learning. Inverse "MDP" models $p(aa'a''|ss''')$ can be used to answer them. In the traditional "forward" view, transition "matrix" $p(s'|sa)$ and policy $\pi(a|s)$ uniquely determine "everything": the whole dynamics $p(as'a's''a''...|s)$, and with it, the action-conditional state process $p(s's''...|saa'a'')$, the multi-step inverse models $p(aa'a''...|ss^i)$, etc. If the latter is our primary concern, a natural question, analogous to the forward case is to which extent 1-step inverse model $p(a|ss')$ plus policy $\pi(a|s)$ determine the multi-step inverse models or even the whole dynamics. In other words, can forward models be inferred from inverse models or even be side-stepped. This work addresses this question and variations thereof, and also whether there are efficient decision/inference algorithms for this.

### Keywords

inverse models; reinforcement learning; causality; theory; multi-step models; planning.

## 1   Introduction

Consider an MDP with actions $a \in \{0,..,k-1\}$ and states $s \in \{1,...,d\}$. Rewards play no role in our analysis, so *controlled Markov process* [DY79] or *conditional Markov chain* may be a more apt naming. Transition "matrix" $p(s'|sa)$ (*"Forward model"*) and policy $\pi(a|s)$ uniquely determine the whole *dynamics*

$$p(as'a's''a''...|s) = \pi(a|s)\cdot p(s'|sa)\cdot \pi(a'|s')\cdot p(s''|s'a')\cdot... \qquad (1)$$

and also determines the action-conditional state process ("Multi-Step Forward Model"):

$$p(s's''...|saa'a'') = p(as'a's''a''...|s)/\sum_{s's''...} p(as'a's''a''...|s) \qquad (2)$$

Here we consider *Inverse Model* $p(a|ss')$ and *Multi-Step Inverse Models* $p(aa'a''...|ss's''s'''...)$ and $p(a|ss^i)$ and variations thereof. Inverse MDP models should not be confused with inverse reinforcement learning [AD21], which infers rewards, which play no role here.

**Motivation.** One motivation to consider inverse models is causal inference: An inverse model captures the likelihood that an action $a$ was the cause of the transition from state $s$ to state $s'$. A multi-step inverse model captures the likelihood that a first action $a$ or action sequence $aa'...a^{i-1}$ was the cause of the state sequence $ss'...s^i$ or the cause of the transition from state $s$ to state $s^i$. The latter is the primary goal in (automatic/stochastic) planning [HSHB99]: to find an action sequence that leads to a desired goal state $s^i = s_{\text{goal}}$. The shortest path, i.e. smallest $i$, that reaches $s_{\text{goal}}$ (with high probability in the stochastic case) can easily be found via a trivial search over $i = 1,2,3,...$ if the fixed-$i$ planning problem can be solved efficiently.

Another machine-learning motivation is that inverse models may be substantially smaller than forward models. For instance, an action-independent Markov process $p(s'|sa) = p(s'|s)$ may be very complex for large $d$, but for a state-independent (known) policy $\pi(a|s) = \pi(a)$, the inverse model $p(aa'...|s..s''..) = \pi(a)\pi(a')...$ is trivial (and known). Of course this extreme case is uninteresting, but a partial similar simplification happens if state $s$ decomposes into $s = (\dot{s}, \ddot{s})$ [EMK$^+$22]. In this case, if the forward model $p(s'|sa)$ factors into a (simple) controlled $p(\dot{s}'|\dot{s}a)$ and (complex) uncontrolled $p(\ddot{s}'|\ddot{s})$, and the policy $\pi(a|s) = \pi(a|\dot{s})$ only depends on (small) $\dot{s}$, then $p(aa'...|s..s''..) = p(aa'...|\dot{s}..\dot{s}''..)$ is independent of (large) $\ddot{s}$. Note that this simplification happens "automatically". We do not need to know the factorization structure, say $(\dot{s}, \ddot{s}) = f(s)$ for some unknown $f$. Appendix B contains a bit of practical context/motivation/application.

**Main questions.**

> *The main question we consider here is:*
> *to which extent do inverse model $p(a|ss')$ plus policy $\pi(a|s)$*
> *determine the multi-step inverse model or even the whole dynamics.*

For instance, do $p(a|ss')$ plus $\pi(a|s)$ determine

  (i) the full dynamics (1),
  (ii) the full dynamics, if also $p(aa'|ss'')$ is provided,
  (iii) the multi-step inverse model $p(aa'...|ss^i)$ (or $p(aa'...|ss's''...)$),
  (iv) the multi-step inverse model $p(aa'...|ss^i)$ (or $p(aa'...|ss's'')$), if also $p(a|ss'')$ is provided,
  (v) just the initial action $p(a|ss'')$ from just final state $s''$,
  (vi) $p(a|ss^i)$ if also $p(a|ss'')$ is provided,

and variations thereof? Also, is there an efficient algorithm that can decide whether the solution is unique and/or computes any or all of them?

Unlike in the "forward" case (1), the answer to all these questions is 'complicated' and 'sometimes'. For instance, (i) is true iff $k \geq d$ and $p(s'|sa)$ has full rank. (ii) seems true for "most" transition matrices. (iii-vi) can fail, but (iv) and (vi) seem to hold for interesting cases. In some situations there are efficient algorithms which sometimes work.

**Related work.** There is of course abundant literature on causal reasoning in general [PGJ16], and in the modern context of Deep Learning in particular [OKD$^+$21], but to the best of our knowledge, the setup and questions we are asking are novel, at least in this generality and rigor.

A special case of our setup is considered in [EMK$^+$22]. The authors consider Exogenous Block MDPs (EX-BMDPs) which correspond to the motivating decomposition example above, and formalized in Section 3 as tensor-product MDPs. Additionally they assume episodic MDPs with near-deterministic dynamics. Their PPE algorithm finds action sequences of high inverse probability $p(aa'...a^{i-1}|ss^i)$ in polynomial time in $\dot{s}$ rather than $s$, while our aim is to infer higher- from lower-step inverse models for general MDPs.

In the context of Deep Learning, there is ample empirical work that would benefit from a positive answer to our main question: Variational Intrinsic Control [GRW17] and Diversity is All You Need [EGIL18] are representative of a broad class of methods that learn diverse options (policies / action sequences) that are inferrable from their effects on the environment. This relies on inverse modelling, as their mutual information objective is decomposed into maximizing skill/policy entropy and minimizing the entropy of an inverse model:

$$I(s^i; a...a^{i-1}|s) \equiv H(a...a^{i-1}|s) - H(aa'...a^{i-1}|ss^i)$$

This is akin to finding all action sequences of sufficiently high probability $p(aa'...a^{i-1}|ss^i)$, or all skills when the policy space is captured by an auxiliary variable $p(z|ss^i)$. The EDDICT algorithm [HDB$^+$21] also maximizes this objective, and parameterizes the requisite inverse models such that they yield forward predictions, but as detailed in Section 4 its unlikely that such models would yield optimal multi-step inverse predictions in general. Dynamics-Aware Unsupervised Discovery of Skills [SGL$^+$19] decomposes the mutual information in the opposite direction, so as to avoid learning an inverse model and instead relies on a conventional forward model. Uniting all of the above mentioned methods is that the action sequence/skill horizon $i$ must be fixed a priori. Inferring long horizon

inverse models from shorter ones (the topic of the present work) would allow all of these methods to circumvent this constraint.

A second stream of empirical work uses single-step inverse models for representation learning [BEP$^+$18]. Agent57 is arguably the most prominent of these methods [BPK$^+$20], and therein the authors note that this choice of representation limits the generality of their approach, as multi-step effects can be aliased over. Despite this being a known limitation, multi-step inverse models are not used as they are too cumbersome to effectively learn online. A positive result to our questions (iii) or (iv) would allow such methods to leverage multi-step inverse predictions despite only learning a single-step model.

These two beneficiaries of improvements to the construction of multi-step inverse models (filtering action sequences and state abstraction) dovetail into potential benefits for a broad range of planning algorithms. Exploiting this relationship between the questions addressed here and planning problems is left to future work, but we sketch out the motivation more fully in Section B.

**Contents.** In Section 2 we will formalize questions (i)-(vi) in matrix/tensor notation. Section 3 gives a first probe into these questions by considering various degenerate cases. In Section 4 we study the solvability and uniqueness questions (i),(iii),(v), when only $B^a$ is given, i.e. the case $i = 1$, in preparation for and showing the necessity of considering $i > 1$. In Section 5 we provide a polynomial-time algorithm via linear relaxation that works under certain conditions. Section 6 provides some validation experiments on toy domains. Section 7 concludes, followed by references. Appendices A-R contain a list of notation, more motivation, counter-examples, experiments, and more.

## 2 Problem Formalization and Preliminaries

We now formalize our questions (i)-(vi) from the introduction, and for this purpose introduce some useful matrix notation. We are not aware of prior work addressing these questions, so quite some ground-work to suitably formalize the various question is needed, and many little results are derived or mentioned in passing to give better insight into the structure of the problem. To avoid clutter, we will not constantly point out edge cases or domain constraints. For instance quantities that represent probabilities are obviously non-negative and sum to one. The reader worried about divisions by $0$ here and there should best assume that all probabilities are strictly positive, but most considerations and results naturally generalize with some care, e.g. by adding "almost surely" w.r.t. to the joint distribution (1). Appendix Q contains a proper treatment of 0/0.

**Notation.** Capital letters $B,D,I,M,W,...$ are used for $d \times d$ matrices over $[0;1] \subset \mathbb{R}$ and tensors by adding further upper indices, e.g. $M_{\cdot \cdot}^{\cdot}$ is an order-3 tensor, and $M_{\cdot}^a$ a matrix for each $a \in \{0 : k-1\} := \{0,...,k-1\}$, and $A,C,V,...$ are other tensors. We define Id to be the identity (eye) matrix $\text{Id}_{ss'} := \delta_{ss'} := [\![s = s']\!] \ \forall s,s' \in \{1 : d\}$, and $I$ to be the all-one matrix $I_{ss'} = 1 \ \forall ss'$. We drop all-quantifiers $\forall s,s',...$ if clear from context. Let $\odot$ denote element-wise (Hadamard) multiplication ($[A \odot B]_{ss'} = A_{ss'} B_{ss'}$), and similarly $\oslash$, while (no) $\cdot$ represents (conventional) matrix multiplication and has operator precedence over $\odot$ and $\oslash$. Matrices form a ring under conventional $(+,\cdot)$ and a commutative ring under $(+,\odot)$, but $(A \cdot B) \odot C \neq A \cdot (B \odot C)$. A diagonal matrix $D$ has the property $D = D \odot \text{Id}$, i.e. $D_{ss'} = D_{ss}[\![s = s']\!]$. $V := I \cdot D$ is a matrix with $D_{ss}$ in the whole of column $s$ ($V_{ss'} = V_{*s'} = D_{s's'}$). Note that $A \cdot D = A \odot V$ ($[A \cdot D]_{ss''} = \sum_{s'} A_{ss'} D_{s's''} = A_{ss''} D_{s''s''} = A_{ss''} V_{*s''} = [A \odot V]_{ss''}$). Similar left-right reversed identities hold. $\perp$ denotes 'undefined'. See Appendix A for a full List of Notation.

**Matrix/tensor formalization.** We define $M_{ss'}^a := p(as'|s) = \pi(a|s)p(s'|sa)$. Marginalizing out the action, gives $p(s'|s) = \sum_a p(as'|s) = \sum_a M_{ss'}^a =: M_{ss'}^+$. Marginalizing out the next-state, gives back $\pi(a|s) = \sum_{s'} p(as'|s) = \sum_{s'} M_{ss'}^a =: M_{sa}^+$. For instance, the multi-step dynamics can be written as

$$p(as'a's''...|s) = p(as'|s) \cdot p(a's''|a') \cdot ... = M_{ss'}^a M_{s's''}^{a'}...$$

Marginalizing out the intermediate states gives

$$p(aa'...a^{i-1}s^i|s) = [M^a \cdot M^{a'}...\cdot M^{a^{i-1}}]_{ss^i}$$

The inverse MDP model can then be expressed as

$$B^a_{ss'} := p(a|ss') = p(as'|s)/p(s'|s) = M^a_{ss'}/M^+_{ss'} = [M^a \oslash M^+]_{ss'}$$

The multi-step inverse model given the whole state sequence becomes

$$p(aa'...|ss's''...) = \frac{p(as'|s)p(a's''|s')...}{p(s'|s)p(s''|s')...} = \frac{M^a_{ss'}M^{a'}_{s's''}...}{M^+_{ss'}M^+_{s's''}...} = p(a|ss')p(a'|s's'')... \qquad (3)$$

and can easily be computed from the 1-step inverse models. To answer the primary question: which action sequence can lead to (desired) state $s^i$ from state $s$, we need to marginalize out $s'...s^{i-1}$. For instance, the two-step inverse model from $s$ to $s''$ with $s'$ marginalized out becomes

$$B^{aa'}_{ss''} := p(aa'|ss'') = \frac{\sum_{s'} M^a_{ss'}M^{a'}_{s's''}}{\sum_{s'} M^+_{ss'}M^+_{s's''}} = [M^a \cdot M^{a'} \oslash (M^+)^2]_{ss''} \qquad (4)$$

Note that unlike the forward case, $B^{aa'} \neq B^a \cdot B^{a'}$, which is responsible for all the problems we will face. Also $B^{a+} \neq B^a$ but $B^+ = 1 = B^{++}$. We always use brackets to denote and disambiguate (matrix) powers $()^2$ from upper indices $M^a$. The initial-action 2-step (and similarly $i$-step) inverse models follow from further marginalizing $a'a''...$:

$$B^{a+}_{ss''} = p(a|ss'') = [M^a M^+ \oslash (M^+)^2]_{ss''},$$
$$B^{a+^{i-1}}_{ss^i} = p(a|ss^i) = [M^a (M^+)^{i-1} \oslash (M^+)^i]_{ss^i} \qquad (5)$$

With this notation, questions (i-vi) in the introduction can formally be written as

 (i) Can $M$ be inferred from $B^a := M^a \oslash M^+$?
 (ii) Can $M$ be inferred from $B^a$ and $B^{aa'} := M^a M^{a'} \oslash (M^+)^2$?
 (iii) Can $B^{aa'...a^i} := M^a M^{a'}...M^{a^i} \oslash (M^+)^i$ be inferred from $B^a$?
 (iv) Can $B^{aa'...a^i}$ be inferred from $B^a$ and $B^{aa'}$?
 (v) Can $B^{a+} := M^a M^+ \oslash (M^+)^2$ be inferred from $B^a$?
 (vi) Can $B^{a++} := M^a (M^+)^2 \oslash (M^+)^3$ be inferred from $B^a$ and $B^{a+}$?

Each question comes in two versions, given also $\pi$, or not knowing $\pi$. We mainly consider the former version, i.e. knowing $M^a_{s+}$:

$$\text{Constraint on } M \text{ for known } \pi: \quad M^a_{s+} = \pi(a|s) \quad \text{and in particular} \quad M^+_{s+} = 1 \qquad (6)$$

Questions (i)-(vi) also have multiple variations:

 (I) Assume some arbitrary $B^a$ (and $B^{aa'}$) is given, but not defined via $M$.
       Is there no, exactly one, or multiple $M$ consistent with these $B$?
 (II) Is there an efficient algorithm that can decide the previous question?
 (III) Is there an efficient algorithm that can compute any/all solutions if one/many exist, and halts/loops if not (4 non-trivial combinations of /).
 (IV) Can we efficiently determine the "number" of solutions,
       e.g. the dimension of the variety formed by the set of all solutions.

**Formulation of the uniqueness questions.** Abstractly, these questions ask whether $M$ (in case of (i-ii)) or $g(M)$ for some function $g$ (in case of (iii-vi)) can be inferred from some other function $f(M)$. Let us define another MDP $q(s'|sa)$ with same policy $\pi(s|a)$ and shorthand

$$W^a_{ss'} := \pi(a|s)q(s'|sa)$$

(In applications, $B^a$ would be learned from data, and $W$ or $B^{aa'\cdots}$ inferred from $B^a$ in the hope that $W \approx M$.) One way to rephrase the questions is whether $f(M) = f(W)$ implies $M = W$ or $g(M) = g(W)$ for all (or most or some) $M$ and $W$. The condition that $\pi$ is the same for $p$ and $q$, translates to

$$\text{Constraint on } M \text{ and } W: \quad M^a_{s+} = \pi(a|s) = W^a_{s+} \quad \text{and in particular} \quad M^+_{s+} = 1 = W^+_{s+} \qquad (7)$$

155 We name the two most interesting equation versions as follows:

$$\text{EqIM}(ia)\colon \quad B^{aa'...a^i} := M^a M^{a'}...M^{a^i} \oslash (M^+)^i \stackrel{?}{=} W^a W^{a'}...W^{a^i} \oslash (W^+)^i \qquad (8)$$

$$\text{EqIM}(i+)\colon \quad B^{a+...+} := \quad M^a (M^+)^{i-1} \oslash (M^+)^i \stackrel{?}{=} W^a (W^+)^{i-1} \oslash (W^+)^i \qquad (9)$$

156 We allow $M_{ss'}^+ = 0$ and keep probabilistic convention that $p(a|ss') = \pi(a|s)p(s'|sa)/p(s'|s)$ is
157 undefined iff $p(s'|s) = 0$ (see end of Appendix J and Appendix Q for more discussion). Formally,
158 $B_{ss'}^a = \perp = 0/0$ iff $M_{ss'}^+ = 0$, also $W_{ss'}^+ = 0$ iff $M_{ss'}^+ = 0$, and similarly for larger $i$.

## 3 Degenerative Cases

160 To get some feeling about why these questions are so more intricate than analogous ones in for-
161 ward models, we consider some simple examples and special cases first, with details provided in
162 Appendix D. Some further special cases (deterministic planning, deterministic reachability, and
163 deterministic inverse models) are considered in Appendix E. There is a strong relationship between
164 the examples violating (i,iii,v) and counter-examples to seemingly different conjectures found in
165 related work. See Section C for details.

166 It is easy to see that e.g. $M^0 = \frac{1}{4}\begin{pmatrix} 0 & 2 \\ 1 & 1 \end{pmatrix}$, $M^1 = \frac{1}{4}\begin{pmatrix} 2 & 0 \\ 1 & 1 \end{pmatrix}$, $W^0 = \frac{1}{2}\begin{pmatrix} 0 & 1 \\ 1 & 0 \end{pmatrix}$, $W^1 = \frac{1}{2}\begin{pmatrix} 1 & 0 \\ 1 & 0 \end{pmatrix}$ satisfy EqIM(1) but
167 violate EqIM(2), which means that the 1-step inverse model $B^a$ does not always uniquely determine
168 the 2-step inverse model $B^{aa'}$, i.e. (i,iii,v) can fail. $M = W$ trivially implies $g(M) = g(W)$, which
169 means that if (i) is true, then trivially also (iii&v), and if (ii) is true, then trivially also (iv&vi). If $M_{ss'}^a$
170 is independent $a$ or $s'$ or $M_{ss'}^a = M_{ss'}\pi_a$, then $B^{aa'a''...} = k^{-i}$ is independent $M$, so any $W \neq M$
171 leads to the same $B$, which shows that (i) and (ii) and higher order analogues can fail. If $M$ and $W$
172 are independent $s$, then EqIM(1) actually implies EqIM($i$)$\forall i$. Since there are such $M \neq W$ satisfying
173 EqIM(1), this constitutes another failure case of (i) and (ii). For block-diagonal $M = \begin{pmatrix} \dot{M} & 0 \\ 0 & \dot{M} \end{pmatrix}$ and
174 $W = \begin{pmatrix} \dot{W} & 0 \\ 0 & \dot{W} \end{pmatrix}$, all operations $(+-\times/\odot\oslash)$ preserve the block structure, so the above degenerative
175 cases can be combined, one for the upper-left block and another for the lower-right block. The most
176 interesting special case is as follows:

177 **Tensor-product $M$ and $W$.** Let $[\dot{M}\otimes\ddot{M}]_{ss'} := \dot{M}_{\dot{s}\dot{s}'}\ddot{M}_{\ddot{s}\ddot{s}'}$ with $s := (\dot{s},\ddot{s})$ and $s' := (\dot{s}',\ddot{s}')$ be the ten-
178 sor product of $\dot{M}$ and $\ddot{M}$ (not to be confused with the element-wise product $\odot$). Assume $M^a = \dot{M}^a \otimes$
179 $\ddot{M}$, where the second factor is action-independent. In this case, $M^a M^{a'}... = (\dot{M}^a \dot{M}^{a'}...)\otimes(\ddot{M}\ddot{M}...)$,
180 and similarly if $a, a',...$ is replaced by $+$, hence $M^a M^{a'}...M^{a^i} \oslash (M^+)^i = \dot{M}^a \dot{M}^{a'}...\dot{M}^{a^i} \oslash (\dot{M}^+)^i$
181 is independent of $\ddot{M}$, and similarly for $W^a = \dot{W}^a \otimes \ddot{W}$. That means, EqIM($i$) hold if $\dot{M}^a = \dot{W}^a$,
182 whatever $\ddot{M}$ and $\ddot{W}$ are. This formalizes our motivating example that if some part of the state ($\ddot{s}$)
183 is not controlled (by $a$) and the dynamics factorizes ($p(s'|sa) = p(\dot{s}'|\dot{s}a)p(\ddot{s}'|\ddot{s})$) and the policy is
184 independent $\ddot{s}$ ($\pi(a|s) = \pi(a|\dot{s})$), then the multi-step inverse models (3-5) become much simpler than
185 the forward model (2), namely independent $\ddot{s}$. This case has been studied in [EMK$^+$22] for episodic
186 near-deterministic $M$.

## 4 (Non)Uniqueness of Inverse MDP Models

188 We will now consider EqIM(1) and EqIM(2). We first provide a dimensional analysis which gives
189 some insight and tentative answers about the solution space for $W$ (given $B$ or $M$): No, one, finitely
190 many, or a polynomial variety (of some dimension) of solutions. We then consider EqIM(1) only
191 and characterize $M$ and $W$ for which it holds. This will be used to provide an algorithm that can
192 determine a (and in some sense all) solution for $W$ and hence $B^{aa'...}$, given only $B^a$. EqIM(1)
193 is quite simple, since it is effectively linear, but EqIM(2) is quadratic in $W$, which is where the
194 difficulties start.

195 **Dimensional analysis / counting solutions.** Assume $k \leq d$ and $B^\cdot$ or $M^\cdot$ are given. The $kd^2$
196 equations EqIM(1) in $W$ constitute $(k-1)d^2$ (linear) constraints on (the $kd^2$ real entries in) $W$. It's
197 only $(k-1)d^2$, since summing over $a$ gives $d^2$ vacuous equations $B^+ = 1 = W^+ \oslash W^+$. There are
198 $kd$ further (linear) constraints $W_{s+}^a = \pi(a|s)$. Assuming no further (missed/accidental) redundancies,
199 this leads to a $kd^2 - (k-1)d^2 - kd = d(d-k)$ dimensional (linear) solution space for $W$. This is

consistent with the algorithm below inferring $B^{aa'}$ from $B^a$ if all $B^a$ have full rank. Hence the set of solutions for $B^{aa'}$ forms a polynomial variety of dimension at least $d(d-k)$.

If also $B^{a+}$ is given, EqIM(2+) provides $(k-1)d^2$ further (quadratic) constraints (EqIM($ia$) even provides $(k^i-1)d^2$ constraints). Since $d(d-k)<(k-1)d^2$, this now gives an over-determined system which generally has no solution. But by assumption, $M$ is a solution, which gives hope that there may be only one or a finite number of solutions.

We can use the $kd+(k-1)d^2$ linear equations to eliminate this number of variables in $W$, which leaves $(k-1)d^2$ quadratic equations, now in only $d(d-k)$ variables, and no further equality constraints. By Bézout's bound [FW89], such a System of Quadratic Equations (SQE), either has a continuum number of solutions (as in the counter-example of Appendix K) or at most $2^{d(d-k)}$ solutions (as possibly in the counter-example in Appendix J). Multiple discrete solutions are often caused by symmetries, so for random $B^a$ and $B^{a+}$ consistent with $M$, the solution may indeed be unique.

**Inferring *some* $B^{aa'}$ from $B^a$.** Even if $B^a$ does not uniquely determine $B^{aa'}$, we can ask for an *algorithm* inferring *some* consistent $B^{aa'}$ from $B^a$. Indeed this was our primary goal before realizing that the answer is not always unique. We know that $B^a = W^a \oslash W^+$ for *some* $W$. This implies $W^a = B^a \odot W^+$. So $W^a = B^a \odot J$ for some $J$ independent $a$. We need to ensure proper normalization $W^a_{s+} = \pi(a|s)$, i.e. $[B^a \odot J]_{s+} = \pi(a|s)$. This leads to the following algorithm to produce some (and indeed all) $B^{aa'}$:

- Given inverse 1-step model $B^a_{ss'} := p(a|ss')$ and policy $\pi(a|s)$
- For each $s$, choose *some* $d$-vector $J_{s\cdot}$
  satisfying the $k$ linear equations $\sum_{s'} B^a_{ss'} J_{ss'} = \pi(a|s)$
- Compute forward model $W^a := B^a \odot J$
- Compute 2-step inverse model $B^{aa'} := W^a W^{a'} \oslash (W^+)^2$
- Then $p(aa'|ss'') \equiv B^{aa'}_{ss''}$ is *some* solution.

If for every $s$, matrix $B^{\cdot}_{s\cdot}$ has rank $d$, then $B^{aa'}$ is unique. The equations have no solution *iff* $B$ is invalid in the sense that no underlying MDP $M$ could have produced such $B$. This can only happen for $k>d$, i.e. $B$ based on $M$ have some intrinsic constraints beyond $B^+=1$ for $k>d$. For instance $B^0 = \frac{1}{2}\begin{pmatrix} 1 & 0 \\ 1 & 0 \end{pmatrix}$, $B^1 = \frac{1}{2}\begin{pmatrix} 0 & 1 \\ 0 & 1 \end{pmatrix}$, $B^2 = \frac{1}{2}\begin{pmatrix} 1 & 1 \\ 1 & 1 \end{pmatrix}$ is inconsistent with $\pi(a|s) = \frac{1}{3}$. For unknown $\pi$, any $J$ with $J_{s+} = 1$ will do. In general, the valid $J$ span a linear subspace, but the set of all consistent $B^{aa'}$ forms an algebraic variety of equal or lower dimension. $B^{aa'}$ may even be unique even if $J$ and $W$ are not (see Section 3). Noting that the ranks of $M^{\cdot}_{s\cdot}$ and $W^{\cdot}_{s\cdot}$ are the same, this gives the precise conditions under which (i) is true:

**Proposition 1 (Conditions under which (i) is true)**

$$M^a \oslash M^+ = W^a \oslash W^+ \text{ implies } M = W \quad \text{iff} \quad M^{\cdot}_{s\cdot} \text{ has rank} \geq d \text{ for every } s.$$

For this to be possible at all, we need $k \geq d$, i.e. more actions than states. This is typically not the most interesting regime. See Appendix F for an alternative derivation of this result without an intermediary algorithm.

We will next show that EqIM(2) removes this limitation, but we do not know of a general and efficient *algorithm* for inferring (some) $B^{aa'a''}$ from $B^a$ and $B^{aa'}$. We cannot even rule out that finding approximate solutions is NP-hard.

**(Non)Uniqueness of Inverse MDP Models for $i \geq 2$.** Above we have established that $B^a$ does not uniquely determine $B^{aa'}$ for the interesting regime of $k<d$. From the dimensional analysis, providing 2-step inverse model $B^{aa'}$ in addition, has the potential of uniquely determining forward model $W$ and/or multi-step inverse models $B^{aa'a''\cdots}$. We have numerically verified that this is indeed the case for $B^a$ and $B^{aa'}$ based on random $M^a$. A more detailed analysis of the linear/quadratic structure of the problem is provided in Appendix G and a rank analyses in Appendices H and R. Unfortunately, even providing $B^a$ *and* $B^{aa'}$ does not always uniquely determine $M^a$, nor higher $B$, and (ii,iv,vi) fail for some $M^a$. Furthermore this remains true for higher $i$-versions, i.e. even EqIM(1)...EqIM($i$) do not always uniquely determine EqIM($i+1$). We provide (potential) counter-examples in Appendices I and J, but they involve "bad" 0/0. We discuss what this means at the end of Appendix J. We provide a

248 fully satisfactory counter-example in Appendix K. If the solution is not unique, the set of solutions
249 forms a polynomial variety. Its (local) dimension measures the "number" of other solutions (in a
250 neighborhood). In Appendix R we provide explicit expressions for the tangent spaces from which
251 these dimension can efficiently be calculated.

## 5  Linear Relaxation

253 In Section 4 we provided an algorithm if only $B^a$ is given. Here we consider the $i > 1$ case, and
254 derive an algorithm for $k^i \geq d$, provided the solution is unique and further conditions on $B$ are met.
255 That is, we require $i \geq \log_k(d)$, which is greater than the minimum necessary in theory $i = 2$ from
256 the dimensional analysis. E.g. for $i = 1$ we recover $k \geq d$, and $i = 2$ improves this to $k \geq \sqrt{d}$, and
257 $i = \lceil \log_2(d) \rceil$ works for all $k$.

258 **Recursive formulation.**   From EqIM(1) we know that $W^a = B^a \odot W^+$. Plugging this into EqIM($ia$)
259 and abbreviating $a^{:i} := aa'...a^i$ and $a^{<i} := aa'...a^{i-1}$ and $j := i+1$, this gives

$$B^{a^{:i}} \odot (W^+)^i = (B^a \odot W^+) \cdot ... \cdot (B^{a^i} \odot W^+) \tag{10}$$

260 If we plug EqIM($(i-1)a$) into EqIM($ia$) and abbreviate $V := (W^+)^{i-1}$ this simplifies to

$$B^{a^{:i}} \odot (V \cdot W^+) = (B^{a^{<i}} \odot V) \cdot (B^{a^i} \odot W^+)$$

261 which written out becomes

$$\sum_{s^i} B^{a^{:i}}_{ss^j} V_{ss^i} W^+_{s^i s^j} = \sum_{s^i} B^{a^{<i}}_{ss^i} V_{ss^i} B^{a^i}_{s^i s^j} W^+_{s^i s^j} \tag{11}$$

262 **Linear relaxation.**   We can consider a linear relaxation of this System of Polynomial Equations
263 (SPE) by introducing new variables $U_{ss^i s^j}$ (aiming at $U_{ss^i s^j} = V_{ss^i} W^+_{s^i s^j}$):

$$\sum_{s^i} A^{a^{:i}}_{ss^i s^j} U_{ss^i s^j} = 0 \quad \text{with} \quad A^{a^{:i}}_{ss^i s^j} := B^{a^{:i}}_{ss^j} - B^{a^{<i}}_{ss^i} B^{a^i}_{s^i s^j} \tag{12}$$

264 These are $k^i d^2$ potentially independent linear equations in $d^3$ unknowns $U$. The solution can only be
265 unique if $k^i \geq d$. For random $B$, for each fixed $(s, s^j)$, the $k^i \times d$ matrix $A^{\cdots}_{s \cdot s^j}$ has indeed full rank
266 $\min\{k^i, d\} \geq d$, hence $U_{ss^i s^j} \equiv 0$ is the only solution. This is inconsistent with the constraints (7),
267 and hence shows that (unrestricted random) $B$ do not come from some $M$. This makes the validity of
268 the $B$'s sometimes semi-decidable in time $O(d^4(d + k^i))$ or typically/randomized time $O(d^5)$. For
269 the $B$'s originating from some $M$, $\hat{U}_{ss^i s^j} = (M^+)^{i-1}_{ss^i} M^+_{s^i s^j}$ solves (12). Since for different $ss^j$ the
270 equations in (12) are independent, $U_{ss^i s^j} := \hat{U}_{ss^i s^j} K_{ss^j}$ also solves (12) for any $K$. In other words,
271 the rank of $A^{\cdots}_{s \cdot s^j}$ is bounded by $\min\{k^i, d-1\}$, and achieved e.g. for random matrices $B$ *consistent*
272 with $M$. Since the solution is not unique, for many solutions $U$ there will be no $W^+$ satisfying
273 $U_{ss^i s^j} = (W^+)^{i-1}_{ss^i} W^+_{s^i s^j}$, not to speak of $M^+$, even if the original problem (10)+(7) has a unique
274 solution.

275 **Unique solution by lifted constraints.**   So we must (and at least for random $M$ can) make the
276 solution unique by taking into account the linear constraints (7). Applying them to $s \rightsquigarrow s^i, s' \rightsquigarrow s^j, a \rightsquigarrow$
277 $a^i$ and multiplying from the left with $V_{ss^i}$ and using $V_{ss^i} = U_{ss^i +}$ we lift them to

$$\sum_{s^j} B^{a^i}_{s^i s^j} U_{ss^i s^j} = U_{ss^i +} \pi(a^i|s^i) \quad \text{and} \quad U_{s++} = 1 \tag{13}$$

278 These $kd^2 + d$ further linear constraints have the potential to make the solution of (12) unique, i.e.
279 resolve the $d^2$ degeneracy $K_{ss^i}$. If so, we can recover $M^+_{s^i s^j} = W^+_{s^i s^j} = U_{ss^i s^j}/V_{ss^i}$ (and finally
280 $M^a = W^a = B^a \odot W^+$) in polynomial time. It actually suffices to solve (12) and (13) for one fixed
281 $s$, e.g. $s = 1$, which with some care can be done in time $O(d^4)$. In practice, for approximate $B$ one
282 would solve a least-squares problem using all equations or a random projection for speed.

283 **Algorithm.**   Putting pieces together, we have the following algorithm for computing $W^a$ and hence
284 $B^{a^{:j}}$ for all $j$ via EqIM($ja$) from $B^a$ and $B^{a^{<i}}$ and $B^{a^{:i}}$

- Given: Policy $\pi(a|s)$ and for $j-1:=i\geq 2$, inverse $1,i-1,i$-step models
  $B^a_{ss'}=p(a|ss')$ and $B^{a^{<i}}_{ss^i}=p(a^{<i}|ss^i)$ and $B^{a^{:i}}_{ss^j}=p(a^{:i}|ss^j)$
- Do the following calculations for one $s$ (e.g. $s=1$),
  or a few or all $s$ or some random linear combinations of $s$:
- For each $s^j$, let $\hat{U}_{ss^is^j}$ be a solution of (12) with $\hat{U}_{s+s^j}=1$
- If a non-zero solution does not exist, set $\hat{U}_{ss^is^j}=0\ \forall s^i$.
- Optional: If multiple solutions exist, return "$W$ *may* not be unique"
- If $\hat{U}_{s++}=0$, return "$B$ is not consistent with any $M$"
- Solve $\sum_{s^j}C^{a^i}_{ss^is^j}K_{ss^j}=0$ and $K_{s+}=1$ for $K_{s*}$, where $C^{a^i}_{ss^is^j}:=(B^{a^i}_{s^is^j}-\pi(a^i|s^i))\hat{U}_{ss^is^j}$
- If no solution, return "$B$ is not consistent with any $M$"
- Optional: If multiple solutions exist, return "$W$ *may* not be unique"
- $\tilde{U}_{ss^is^j}:=\hat{U}_{ss^is^j}K_{ss^j}$,   $U_{ss^is^j}:=\tilde{U}_{ss^is^j}/\tilde{U}_{s++}$,   $V_{ss^i}:=U_{ss^i+}$,   $W^+_{s^is^j}:=U_{ss^is^j}/V_{ss^i}$
- Optional: If different $s$ lead to different $W^+$ or $V\neq(W^+)^{i-1}$,
  return "$W$ *may* not be unique"
- Return forward model $W^a:=B^a\odot W^+$ and other inverse $B^{\cdots}$ computed via (8)

**Variations that don't work.**  For unknown $\pi$, we only have $d$ lifted constraints $U_{s++}=1$, which are not sufficient to make the solution unique, also resulting in too many solutions for the relinearization trick [CKPS00] to work. The same is true if we had relaxed $U_{ss's^j}=W^+_{ss'}V_{s's^j}$. If we had applied linear relaxation directly to EqIM($ia$), this would have led to order-$i+1$ tensors and require $k\geq d^{1-1/i}$, which is much worse than $k\geq d^{1/i}$ for $i>2$. Including $B^{a^{:j}}$ and EqIM($ja$) for some or all $j<i-1$ is not only unhelpful but even counter-productive.

# 6   Experiments

The algorithm described in Section 5 was motivated by the dimensional analysis and properties of random matrices. Namely, that $A^{\cdots}_{s_{:}sj}$ is likely "full" rank, and thus yielding a unique solution. In order to explore the plausibility of this assumption in practice, we have evaluated the algorithm on a set of toy (but structured) environments. This includes the canonical 'four-rooms' grid-world and samples from the distribution over all grid-worlds of that size. All environments have $k=5$ (local movement on the grid) and $d=24$, thus satisfying the $k\geq d^2$ constraint which permits solving EqIM(2).

**Experiments on naturalistic environments.**  As detailed in Appendix P, for all environments tested the algorithm yielded a unique solution (recovering $M^a$) up to a reasonable level of numerical precision. This remained true even after injecting noise (across several orders of magnitude) into the environmental transition dynamics. This is in contrast to related methods which rely on near-deterministic environments [EMK+22].

This result is non-trivial, as the statistics of these environments differ significantly from those produced by random matrices. For example, grid-world dynamics are both local and sparse, unlike random matrix dynamics which almost always have non-zero probability for all transitions. It remains to be seen whether or not larger-scale environments yield similar results, but it is at least non-obvious what additional environmental properties would break the constraints of the algorithm.

**Experiments illustrating robustness to noise.**  The propositions (and previous experimental result) assume that we know the one and two step inverse models ($B1:=B^a$, $B2:=B^{a+}$) exactly, but in practice these distributions must be estimated from data. Here we investigate the extent to which our algorithm is robust to noise arising from learning.

Rather than committing to a specific learning algorithm, we instead directly inject noise into the true inverse distributions. Figure 2 shows that noise doesn't substantially degrade performance across several orders of magnitude (see Appendix P for details). Additionally, the effect of this noise is substantially diminished as the horizon of the inverse model is increased (from $B1:=B^a$ to $B3:=B^{a++}$). While the is perhaps not surprising, as the entropy of such inverse distributions increases monotonically with the horizon, it still shows that noise is not compounding in a way that renders long-horizon predictions meaningless.

**Experiments on the Tensor-product special case.** As detailed in Section 3, if $M$ factors into two processes $\dot{M}^a \otimes \ddot{M}$, where $\ddot{M}$ is action-independent, then only the complexity of the action-dependent process $\dot{M}^a$ matters for all of our questions. The significance of this special case, as well as the details of environments construction, can be found in Appendix P.

The linear algorithm of Section 4 can (implicitly) output all $W$ and $B2$ consistent with $B1$, and the formulas derived in Appendix R allow to (explicitly) calculate the dimensions of the solution spaces.

In the experiments shown in Figure 3, the environments complexity is systematically varied. The results show that the space of forward dynamics $W$ is always larger than the space of the 2-step inverse models ($B2$). This confirms that inverse models can be simpler than forward models.

# 7 Conclusion

**Summary.** We have shown that the 1-step inverse model $p(a|ss')$ does not uniquely determine the 2-step probabilities $p(a|ss'')$ if there are less actions than states ($k < d$). Even for $k \geq d$, the implication can fail, e.g. if the extra actions are ineffective, but if $p(s'|sa) = M^a_{ss'}$ considered as matrices in $a$ and $s'$ for each $s$ have full rank, the implication holds. Even providing $p(aa'...a^{j-1}|ss^j)$ for all $j < i$ not necessarily determines $p(a|ss^i)$. Since the involved SPE is (heavily) over-determined, we expect the failure cases to be sparse/rare in some sense. For ($B$ based on) random $M$, we provided evidence that $a = 2$ suffices to determine $M$ and hence $p(aa'...|ss's''...)$ from $p(a|ss')$ and $p(a|ss'')$. For low-rank $M$ the implication may fail.

**Open Problems.** Maybe characterizing all $M$ for which EqIM(1) and EqIM(2) uniquely determine $W$ is hopeless, not to speak of finding some or all $W$ in case not. More formally, we can ask the question of whether there exists an efficient algorithm that can decide whether EqIM($i$) has a unique solution.

**Conjecture 2 (NP-hardness)** *Deciding (ii), (iv), (vi) is NP-hard. Deciding whether $B^a$ and $B^{aa'}$ are consistent with some $M$ is also NP-hard. Computing some solution is FNP-hard.*

In Appendix L we provide some weak preliminary evidence, why this problem may be NP-hard. Appendix O contains fully self-contained a few versions of this open problem in their simplest instantiation and most elegant form.

**Discussion.** Given our analysis, we would expect that in practice, $B^a$ and $B^{aa'}$ determines $B^{aa'a''\cdots}$ and $W$ sufficiently well. Sufficiently well in case of $W$ means all and only those aspects of the forward model relevant for the inverse model. Then of course the question remains how to compute the/an answer. While the linear relaxation developed in Section 5 fails for $k < d^{1/i}$ as an exact method, it might still lead to useful approximate solutions [Stu02] without formal guarantees. Indeed, EqIM($ia$) is heavily over-determined for $i \geq 2$, and heuristic solvers often work well in this regime.

Handling non-uniqueness: In practice, the state space is very often infinite, and no finite amount of data will determine even $B^a$ uniquely without further structural assumptions. Neural networks intrinsically restrict the solution space, but this may not suffice for modern over-parametrized deep networks. Aiming for the maximum-entropy distribution consistent with the (constraints from) data is popular, and could make the solution unique, as well as any other optimization constraint.

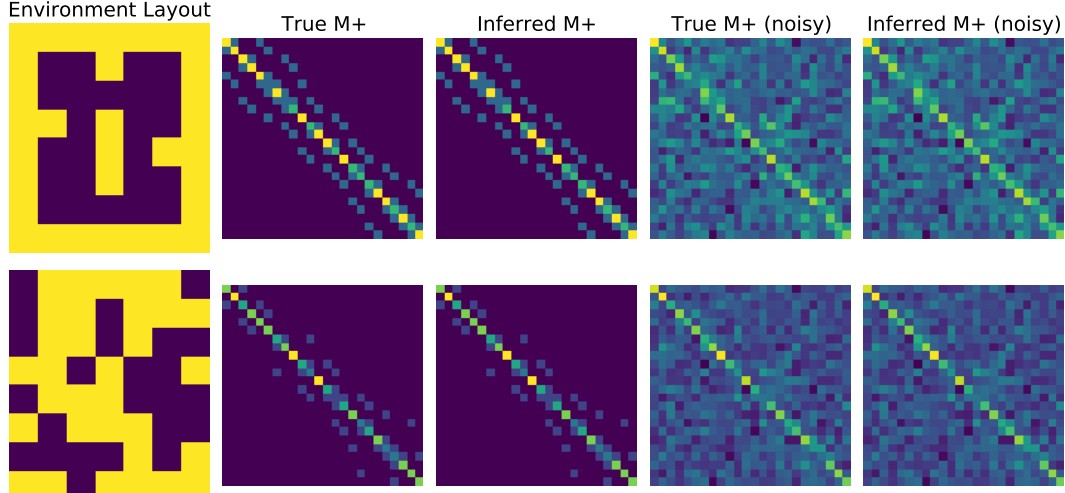

Figure 1: Environments, their transition matrices (i.e. $M^+$) and the matrices inferred by the algorithm (i.e. $W^+$). Results shown on the most and least noisiest variants of each environment. *Top* 'four-rooms' grid-world. *Bottom* One of the randomly generated grid-worlds.

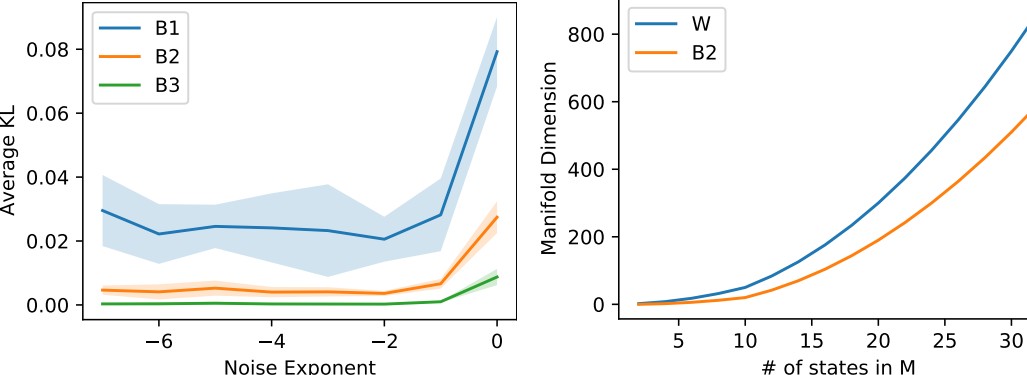

Figure 2: **Noise-induced reconstruction error:** In practice $W$ must be inferred from learned estimates of $B1$ and $B2$. We investigate the effect of the resulting error on the inverse models $(B1,B2,B3)$ recovered from the inferred $W$ in terms of their proximity to the ground truth distributions. At each noise level the algorithm was run on 10 randomly generated grids, with the shaded region representing $\pm 2\sigma$.

Figure 3: **Solution dimensions of $W$ and $B2$ given $B1$:** When the solution to an inverse model $(B2)$ given only $B1$ is not unique, we can characterize the solution space in terms of its manifold dimension. By comparing this to the dimension of that of the inferred forward model $(W)$, we can see that our algorithm has narrowed down the space of inverse models further. If also $B2$ is given, the solution dimension of $W$ reduces from $d_W$ (blue curve) to $d_W - d_B$ (blue minus orange curve).

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

# Inverse MDP Models – Supplementary Material

## A List of Notation

| Symbol | Type | Explanation |
|---|---|---|
| $\perp$ | | undefined |
| $[\![\text{bool}]\!]$ | $\in \{0,1\}$ | =1 if bool=True, =0 if bool=False |
| $\delta_{ss'}$ | $:= [\![ s=s' ]\!]$ | Kronecker delta |
| $d$ | $\in \mathbb{N}$ | number of states |
| $k$ | $\in \mathbb{N}$ | number of actions |
| $i,j$ | $\in \mathbb{N}$ | time index/step |
| $\{i:j\}$ | $\subset \mathbb{Z}$ | Set of integers from $i$ to $j$ (empty if $j < i$) |
| $s,s',...,s^i$ | $\in \{1:d\}$ | state at time step 1,2,...,$i$ |
| $a,a',...,a^i$ | $\in \{0:k-1\}$ | action at time step 1,2,...,$i$ |
| $b,b',...,b^i$ | $\in \{0:k-1\}$ | alternative action at time step 1,2,...,$i$ |
| $a^{:i}$ | $:= aa'...a^i$ | sequence of $i$ actions |
| $a^{<i}$ | $:= aa'...a^{i-1}$ | sequence of $i-1$ actions |
| $\dot{s},\ddot{s}$ | $\in \{1:\dot{d}\}$ | parts of state, usually $s=(\dot{s},\ddot{s})$ |
| $\varepsilon$ | $> 0$ | small number $> 0$ |
| $p(...)$ | $\in [0;1]$ | (conditional) probability distribution over states and actions |
| $\pi(a\vert s)$ | $\in [0;1]$ | policy. Probability of action $a$ in state $s$ |
| $M^a, W^a$ | $\in [0;1]^{d\times d}$ | transition-policy tensor $M^a_{ss'} = p(s'\vert sa) \cdot \pi(a\vert s)$, similarly $W = q$ |
| $B^a$ | $\in [0;1]^{d\times d}$ | inverse 1-step model $B^a_{ss'} = p(a\vert ss')$ for each action $a$ |
| $B^{a++}_{ss'''}$ | $\in [0;1]$ | 3-step first-action inverse model $p(a\vert ss''')$ |
| $J,K,\Delta$ | $\in \mathbb{R}^{d\times d}$ | action-independent $d\times d$ "transition" matrices |
| $^+_{\phantom{+}+}$ | $\cdot^n \to \cdot$ | index summation, e.g. $M^+_{s+} = \sum_{as'} M^a_{ss'}$ |
| $\cdot$ | $(\cdot,\cdot) \to \cdot$ | matrix multiplication: $[AB]_{ss''} = \sum_{s'} A_{ss'} B_{s's''}$ |
| $\odot$ | $(\cdot,\cdot) \to \cdot$ | element-wise multiplication of matrix elements: $[A\odot B]_{ss'} = A_{ss'} B_{ss'}$ |
| $\oslash$ | $(\cdot,\cdot) \to \cdot$ | element-wise division of matrix elements: $[A\oslash B]_{ss'} = A_{ss'}/B_{ss'}$ |
| $\otimes$ | $(\cdot,\cdot) \to \cdot$ | tensor product: $[\dot{M}\otimes\ddot{M}]_{ss'} := \dot{M}_{\dot{s}\dot{s}'}\ddot{M}_{\ddot{s}\ddot{s}'}$ with $s=(\dot{s},\ddot{s})$ and $s'=(\dot{s}',\ddot{s}')$ |

## B Application to Planning

In Section 1, various streams of applied work were highlighted; here we focus on spelling out the overarching impact that compositional inverse models (an affirmative answer to question (iv)) would have for planning problems.

Many forms of planning involve the evaluation of candidate $i$-step action sequences (e.g. model predictive path integral control [WDG$^+$16]). Ideally, all possible action sequences would be evaluated, but as the space of $i$-step action sequences grows exponentially in $i$, this is often intractable.

Access to the $i$-step inverse distribution $p(a...a^i\vert s...s^{i+1})$ allows determining the subset of action sequences that likely reach state $s^{i+1}$ post-execution (e.g. those whose probability is above some threshold). It is often the case that only action sequences that are distinguished in this way are of interest (e.g. goal-reach tasks), thus access to an inverse model of the appropriate horizon allows for filtering candidates. This filtering method is a particularly appealing approach when the cost/reward function is initially unknown and frequently changes, as in [MJR15].

**Motivating Example.** Consider an agent who has control over $\dot{s}$ but not over $\ddot{s}$. For instance a robot equipped with a camera can control its position and orientation, but not the shape and color of objects in its path. The forward model $p(s'\vert as)$ essentially involves modelling the whole observable world. The inverse model $p(a\vert ss')$ on the other hand can ignore inputs that the agent has no control over. Of course in practice, $s$ does not come neatly separated into $\dot{s}$ and $\ddot{s}$, so a (say) deep neural network still has to learn the controllable features, but neither needs to learn nor predict the uncontrollable features (under the factorization assumptions described in Section 3, now in feature space).

If the goal is to navigate from $s$ to $s^i$ in $i$ time steps, and open-loop control suffices as e.g. in (near)-deterministic problems [EMK$^+$22], then action sequences for which $p(aa'...a^{i-1}\vert ss^i)$ is large

are the most likely that caused the transition to $s^i$, hence these sequences are promising candidates for macro actions (temporally extended actions, options) in Reinforcement Learning [SP02, Pre00].

Since the action space is typically much smaller than the state space (the former often finite, the latter often even infinite-dimensional), even learning $p(aa'...a^{i-1}|s...s^i)$ directly for all small $i$ can be feasible and may be more data-efficient than learning the one-step forward model. A closed-loop alternative would be to learn only $p(a|s...s^i)$, find the likely first action $a$ that caused the ultimate transition to $s^i$, then take action $a$, iterate, and store the resulting sequence as an option.

The required sample complexity to learn inverse MDP models for larger $i$ directly from data may grow exponentially in $i$, which is why inferring $i$-step inverse models from 1-step and 2-step inverse models would be useful. The fact that this problem borders NP-hardness probably prevents even powerful transformer models to finding the structure in $p(aa'...a^{i-1}|s...s^i)$ by themselves.

## C  Counter-Examples in Related Work

In Section 3 we presented a counter-example to questions (i,iii,v). Question (i) (i.e. Can $M$ be inferred from $B^a := M^a \oslash M^+$?) has been implicitly addressed in previous work. In [EMK$^+$22, App.A.3] the authors present a counter-example to the claim that a state representation constructed via an inverse model (i.e. two states have the same representation iff they yield the same inverse distribution for all of their possible successor states) is sufficient for representing a set of policies that differentially visit all states. This fails whenever two states are aliased by the inverse model. Technically, as per their Definition 2, this 'policy cover' need only account for all 'endogenous' states. But omit the 'exogenous' states from their counter-example and it can be seen to address our question (i).

Note that this failure of state representation learning implies a negative answer to our question (i), as $W$ would differ from $M$ on these aliased states. Unlike our counter-example, theirs involves deterministic forward dynamics, and therefor buttresses our claims by showing that $M$ cannot always be inferred even in this simpler case. Similar to our counter-example in Section 3, [MHKL20] proposes a stochastic counter-example to inverse modeling for state representation learning.

In general, the transferability of these counter-examples suggests a strong relationship between the literature on using single-step inverse models for state representation learning and using them for inferring the forward model. It is an interesting open question whether or not algorithms for representation learning on the basis of multi-step inverse models (like those put forward in [EMK$^+$22]) might be used to shed light on the questions put forward here and vice versa.

## D  Degenerative Cases - Details

To get some feeling about why these questions are so more intricate than analogous ones in forward models, we consider some simple examples and special cases first. Some further special cases (deterministic planning, deterministic reachability, and deterministic inverse models) are considered in Appendix E.

**Example violating (i,iii,v).**  A specific example for $M$ and $W$ which satisfy EqIM(1) but violate EqIM(2+) and hence EqIM(2a) is as follows:

$$M^0 = \tfrac{1}{4}\begin{pmatrix}0 & 2\\1 & 1\end{pmatrix}, \quad M^1 = \tfrac{1}{4}\begin{pmatrix}2 & 0\\1 & 1\end{pmatrix}, \quad W^0 = \tfrac{1}{2}\begin{pmatrix}0 & 1\\1 & 0\end{pmatrix}, \quad W^1 = \tfrac{1}{2}\begin{pmatrix}1 & 0\\1 & 0\end{pmatrix}$$

which satisfies (7) ($M^a_{s+} = \tfrac{1}{2} = W^a_{s+}$). In this example, $M^+ = \tfrac{1}{2}\begin{pmatrix}1 & 1\\1 & 1\end{pmatrix}$ and $W^+ = \tfrac{1}{2}\begin{pmatrix}1 & 1\\2 & 0\end{pmatrix}$, which shows $M^a \oslash M^+ = W^a \oslash W^+$, except that $W^+_{22} = 0 \neq 1 = M^+_{22}$, hence there is one "dubious" $1 \overset{?}{=} 0/0$ case. A simple calculation shows that EqIM(2+) is violated (w/o any division by 0). The division by 0 can easily avoided by mixing $U^a_{ss'} \equiv \tfrac{1}{4}$ into $M$ and $W$, e.g. $M \leadsto \tfrac{1}{2}(M+U)$ and $W \leadsto \tfrac{1}{2}(W+U)$. This means that the 1-step inverse model $B^a$ does not always uniquely determine the 2-step inverse model $B^{aa'}$, i.e. (i,iii,v) can fail.

**$M = W$.**  This trivially implies $g(M) = g(W)$. This means if (i) is true, then trivially also (iii) and (v), and if (ii) is true, then trivially also (iv) and (vi).

**$M$ and $W$ are independent $a$.** Note that $M_{ss'}^a \equiv p(s'|sa)$ independent $a$ implies $M_{s+}^a$ independent $a$, hence $\pi(a|s) = M_{s+}^a = 1/k$ independent $a$ as well, hence $M^a = \frac{1}{k}M^+$. The latter implies $M^a M^{a'}...M^{a^i} \oslash (M^+)^i = k^{-i}$ is independent $M$ hence is the same as for $W$. Since we can choose $M \neq W$, this shows that (i) and (ii) and higher order analogues fail for these degenerate $M$ and $W$.

**$M$ and $W$ are nearly independent $a$.** The above degeneracy generalizes to $M_{ss'}^a = M_{ss'}\pi_a$ and $W_{ss'}^a = W_{ss'}\pi_a$, i.e. action-independent dynamics, and state-independent actions, which in turn is a special case of the tensor product below (with $s = \ddot{s}$ and $\dot{s} \equiv 0$).

**$M$ and $W$ are independent $s'$.** In this case, $M_{ss'}^a = \frac{1}{d}M_{s+}^a = \frac{1}{d}\pi(a|s) = W_{ss'}^a$, hence is a special case of case $M = W$ above.

**$M$ and $W$ are independent $s$.** In this case, $[M^a M^{a'}]_{ss''} = \sum_{s'} M_{*s'}^a M_{*s''}^{a'} = \pi(a|*)M_{*s''}^{a'}$. Also the policy $\pi(a|s) = M_{s+}^a$ is independent $s$. If we assume EqIM(1), this implies

$$[M^a M^{a'} \oslash (M^+)^2]_{ss''} = \frac{\pi(a|*)M_{*s''}^{a'}}{\pi(+|*)M_{*s''}^+} = \frac{\pi(a|*)W_{*s''}^{a'}}{\pi(+|*)W_{*s''}^+} = [W^a W^{a'} \oslash (W^+)^2]_{ss''}$$

hence EqIM(2) holds and similarly EqIM($i$)$\forall i$. As an example, consider

$$M^0 := \tfrac{1}{2}\begin{pmatrix} 0 & 1 \\ 0 & 1 \end{pmatrix}, \quad M^1 := \tfrac{1}{2}\begin{pmatrix} 1 & 0 \\ 1 & 0 \end{pmatrix}, \quad W^0 := \tfrac{1}{3}\begin{pmatrix} 0 & 1 \\ 0 & 1 \end{pmatrix}, \quad W^1 := \tfrac{2}{3}\begin{pmatrix} 1 & 0 \\ 1 & 0 \end{pmatrix}$$

These $M \neq W$ satisfy EqIM(1) ($M^a \oslash M^+ = 2M^a = W^a \oslash W^+$), hence constitute another failure case of (i) and (ii).

**Block-diagonal $M$ and $W$.** For $M = \begin{pmatrix} \dot{M} & 0 \\ 0 & \dot{M} \end{pmatrix}$ and $W = \begin{pmatrix} \dot{W} & 0 \\ 0 & \dot{W} \end{pmatrix}$, all operations ($+-\times/\odot\oslash$) preserve the block structure, so the above degenerative cases can be combined, one for the upper-left block and another for the lower-right block.

**Tensor-product $M$ and $W$.** Let $[\dot{M} \otimes \ddot{M}]_{ss'} := \dot{M}_{\dot{s}\dot{s}'}\ddot{M}_{\ddot{s}\ddot{s}'}$ with $s := (\dot{s},\ddot{s})$ and $s' := (\dot{s}',\ddot{s}')$ be the tensor product of $\dot{M}$ and $\ddot{M}$ (not to be confused with the element-wise product $\odot$). Assume $M^a = \dot{M}^a \otimes \ddot{M}$, where the second factor is action-independent. In this case, $M^a M^{a'}... = (\dot{M}^a \dot{M}^{a'}...) \otimes (\ddot{M}\ddot{M}...)$, and similarly if $a,a',...$ is replaced by $+$, hence $M^a M^{a'}...M^{a^i} \oslash (M^+)^i = \dot{M}^a \dot{M}^{a'}...\dot{M}^{a^i} \oslash (\dot{M}^+)^i$ is independent of $\ddot{M}$, and similarly for $W^a = \dot{W}^a \otimes \ddot{W}$. That means, EqIM($i$) hold if $\dot{M}^a = \dot{W}^a$, whatever $\ddot{M}$ and $\ddot{W}$ are. This formalizes our motivating example that if some part of the state ($\ddot{s}$) is not controlled (by $a$) and the dynamics factorizes ($p(s'|sa) = p(\dot{s}'|\dot{s}a)p(\ddot{s}'|\ddot{s})$) and the policy is independent $\ddot{s}$ ($\pi(a|s) = \pi(a|\dot{s})$), then the multi-step inverse models (3-5) become much simpler than the forward model (2), namely independent $\ddot{s}$. This case has been studied in [EMK$^+$22] for episodic near-deterministic $M$.

# E  Deterministic Cases

**Deterministic planning / reachability problem.** If we are only interested in finding *some* action sequence $aa'...a^i$ that leads to $s^i$, the problem becomes easy: The only thing that matters is the support of the various matrices, not the numerical values themselves. Since $B_{ss'}^a > 0$ iff $M_{ss'}^a > 0$ (either assuming $M_{ss'}^+ > 0$ or regarding $\perp > 0$ as False), and similarly for higher orders, we can replace $M^a$ by $B^a$ in (iii), and get $B_{ss^{i+1}}^{aa'...a^i} > 0$ iff $[B^a B^{a'}...B^{a^i}]_{ss^{i+1}} > 0$. We could also replace $M^a$ by $G_{ss'}^a := [\![B_{ss'}^a > 0]\!]$, then $[G^a G^{a'}...G^{a^i}]_{ss^{i+1}} > 0$ counts the number of paths of length $i$ from $s$ to $s^{i+1}$ via action sequence $aa'...a^i$, and hence determines whether $s^{i+1}$ can be reached. Similarly $(G^+)^i > 0$ iff there is *some* action sequence that can reach $s^{i+1}$ from $s$. An action $a$ such that $G^a(G^+)^i > 0$ can be chosen as the first action of such a sequence if it exists, and $a',a''...$ can be found the same way by recursion. So this deterministic planning/reachability problem has a "unique" solution, which can be found in time $O(i \cdot d \cdot (d+k))$ (for fixed $s$ and $s^{i+1}$).

**$B$ is deterministic.** Assume $M_{ss'}^a / M_{ss'}^+ =: B_{ss'}^a \in \{0,1,\perp\}$. This is true if and only if $M^a$ has disjoint support for different $a$, i.e. iff $M^a \odot M^b = 0 \, \forall a \neq b$. This in turn means that $B_{ss'}^a = [\![W_{ss'}^a > 0]\!]$ for any and only those $W$ with same support as $M$, and hence also $W^a \odot W^b = 0 \, \forall a \neq b$, which is another failure case of (i). Here we have included the case where *no* action leads from $s$ to $s'$, in which

case $W_{ss'}^+ = 0$ and $B^a$ is undefined ($\perp$). This readily extends to higher orders: If $B^{aa'\cdots} \in \{0, 1, \perp\}$, then $B^{aa'\cdots} = [\![ W^a W^{a'\cdots} \oslash (W^+)^i > 0 ]\!]$ iff $W^a W^{a'}...$ has the same support as $M^a M^{a'}...$ and

$$W^a W^{a'}...W^{a^i} \odot W^b W^{b'}...W^{b^i} = 0 \quad \forall aa'...a^i \neq bb'...b^i \tag{14}$$

Note that $W^a \odot W^b = 0$ does not necessarily imply (14), e.g. for $W^0 = \frac{1}{2}\begin{pmatrix} 1 & 0 \\ 0 & 1 \end{pmatrix}$ and $W^1 = \frac{1}{2}\begin{pmatrix} 0 & 1 \\ 1 & 0 \end{pmatrix}$, $(W^0)^2 = (W^1)^2$. In Appendices I&J&K we construct $W$ such that (14) holds for larger $i$.

## F  Characterizing $M$ and $W$ for which EqIM(1) holds

$$M^a \oslash M^+ = W^a \oslash W^+ \quad \Longleftrightarrow \quad W^a = M^a \odot J \quad \text{with} \quad J := W^+ \oslash M^+$$

That is, $J$ is independent of $a$. Phrased differently

$$\text{For any } M \text{ and } W, \text{ EqIM(1) is satisfied} \quad \textit{iff} \quad W^a \oslash M^a \text{ is independent } a. \tag{15}$$

For a given $M$, this allows to determine all $W$ consistent with EqIM(1), by just multiplying with any $a$-independent $J \geq 0$. Not all $J$ though lead to $W$ consistent with (7). In order to also satisfy (7), $J$ needs to be restricted as follows: With $\Delta_{ss'} := J_{ss'} - 1$, (7) becomes

$$0 \stackrel{!}{=} W_{s+}^a - M_{s+}^a = \sum_{s'} M_{ss'}^a (\Delta_{ss'} + 1) - M_{s+}^a = \sum_{s'} M_{ss'}^a \Delta_{ss'} \tag{16}$$

For each fixed $s$, these are $k$ homogenous linear equations (one for each $a$) in $d$ variables. Given $M$, all and only the $W$ consistent with EqIM(1) *and* (7) can be obtained via $W^a = M^a \odot (1 + \Delta)$ with $\Delta$ satisfying $M_{s\cdot} \Delta_{s\cdot} = 0$.

As a special case, $\Delta = 0$ necessarily if and only if the rank of $M_{s\cdot}$ is $\geq d$ for every $s$. This gives the precise conditions as stated in Proposition 1 under which $(i)$ is true. We will next show that EqIM(2) removes this limitation.

## G  Characterizing $M$ and $W$ for which EqIM(1) and EqIM(2+) hold

From Appendix F we know that the most general Ansatz for $W^a$ satisfying EqIM(1) is $M^a \odot (1 + \Delta)$. Plugging this into (28) and expanding in $\Delta$, we get

$$\begin{aligned}
0 = \ & M^a M^+ \odot (M^+)^2 - M^a M^+ \odot (M^+)^2 \\
& + \ M^a M^+ \odot [M^+(M^+ \odot \Delta) + (M^+ \odot \Delta) \odot M^+] - [(M^a \odot \Delta) M^+ M^a (M^+ \odot \Delta)] \odot (M^+)^2] \\
& + \ M^a M^+ \odot (M^+ \odot \Delta)^2 - (M^a \odot \Delta)(M^+ \odot \Delta) \odot (M^+)^2
\end{aligned}$$

This is a collection of quadratic equations in $\Delta$. The $\Delta$-independent first line is 0. We can write this in canonical form:

$$\begin{aligned}
\Sigma_{kl} A_{ss'',kl}^a \Delta_{kl} = R_{kl}^a(\Delta) \quad \text{with} & \tag{17} \\
A_{ss'',kl}^a := (\Sigma_{s'} M_{ss'}^a M_{s's''}^+)(M_{sk}^+ M_{ks''}^+ \delta_{ls''} + M_{sl}^+ M_{ls''}^+ \delta_{sk} - M_{sk}^a M_{ks''}^+ \delta_{ls''} - M_{sl}^a M_{ls''}^+ \delta_{sk}) & \\
R^a(\Delta) := (M^a \odot \Delta)(M^+ \odot \Delta) \odot (M^+)^2 - M^a M^+ \odot (M^+ \odot \Delta)^2 &
\end{aligned}$$

Let us consider $A^a$ as a $d^2 \times d^2$ matrix for each $a$, $\Delta$ as a vector of length $d^2$, and (wrongly) presume $R^a \equiv 0$ at first. $A^a$ is a sum of 4 terms. The second and fourth terms are block-diagonal matrices ($d$ blocks of size $d \times d$ in the diagonal) due to the $\delta_{sk}$. The first and third terms are scrambled block-diagonal matrices due to the $\delta_{ls''}$, or more precisely, consist of $d \times d$ blocks, each bock being a $d \times d$ diagonal matrix. If $M^a$ has full rank, each of the four terms has full rank $d^2$, but $A^a$ itself can have lower rank, 0-eigenvalues due to some cancellations. Random $M$ apparently achieves the highest rank, but even then, $A^a$ itself has only rank $d(d-1)$.

Actually, $A^a \Delta = 0$ is required to hold for all $a$, so the rank of $A$ as a $kd^2 \times d^2$ matrix may still be $d^2$. But $A^+ \equiv 0$ for $k = 2$ implies $A^0 = -A^1$, hence the rank is still at most $d(d-1)$. $k > 2$ may rectify this, but there is an alternative, which works for all $a$: $\Delta$ also needs to satisfy (16), which can be

rewritten as

$$\sum_{kl} C^a_{s,kl}\Delta_{kl} = 0 \quad \text{with} \quad C^a_{s,kl} := M^a_{sl}\delta_{sk} \tag{18}$$

These give another $kd$ constraints, and apparently often $d$ new ones from random $M$. If we combine $A' := \binom{A^\cdot}{C^\cdot}$, this implies that $A'$ has often rank $d^2$, so $A'\Delta = 0$ can only be satisfied for $\Delta = 0$. For $k=2$, $A^+ = 0$, so inclusion of either $A^0$ or $A^1$ in $A'$ would suffice, but $C^0$ and $C^1$ are potentially independent, so both have to be included.

Let us now return to the real case of $R^a \neq 0$ for full random $M$, hence full-rank $A'$. With $R' := \binom{R^\cdot}{0}$, we need to solve $A'\Delta = R'$. Note that $R' = R'(\Delta)$ is not a constant, but a (homogenous) quadratic function of $\Delta$ itself. Consider any $\Delta = \Theta(\varepsilon)$, then $A'\Delta = \Theta(\varepsilon)$ while $R'(\Delta) = \Theta(\varepsilon^2)$, which is a contradiction for sufficiently small $\varepsilon$ (this argument can be made rigorous). This implies that no $\Delta$ with $0 < ||\Delta|| < \varepsilon$ can satisfy $A'\Delta = R'(\Delta)$. In conclusion,

**Proposition 3 (Random $M$ and full-rank $A'$)**

*If $A'$ has full rank and $W$ is close to $M$, then EqIM(1) and EqIM(2) imply $W = M$.*
*Empirically $A'$ has full rank for random $M$.*

This of course implies EqIM$(i)\forall i$ and also (iv). Globally, i.e. if $W$ is not close to $M$, these implications may not hold.

We have yet to establish sufficient conditions which $M^a$ lead to full-rank $A'$. Empirically, this has been true for random $M^a$, so should hold almost surely if $M$ are sampled uniformly. One might conjecture that full-rank $M^a$ are sufficient, but this is not the case. For instance, if $M^a$ is independent $a$, then $A' \equiv 0$.

**Zero $A$ and $R$ for full-rank $\dot{M}^a$.** We finally we note that $A$ and $R$ can have low rank, indeed $A \equiv 0 \equiv R$ even for $a$-dependent full-rank $M^a$: Consider the example $\dot{M}^a$ from (22) or its generalization (27): First, if for two matrices $M^a$ and $M^{a'}$ only one $s'$ (depending on $s$ and $s''$) contributes to the sum in $M^a M^{a'}$ then $(M^a \odot J)(M^{a'} \odot J) = M^a M^b \odot K$ for some $K$. This makes (19) valid for $M^a := \dot{M}^a$ and $W^a := \dot{M}^a \odot J$ for any $J$, since for $aa' \neq bb'$ both sides are 0 by construction of $\dot{M}^a$ (the $\odot K$ does nothing to it), and are trivially equal for $aa' = bb'$. By summing over $a'bb'$, also (28) is valid for any $J$, hence of course also for $J = 1 + \Delta$ for any $\Delta$. Since (17) is equivalent to (28), (17) holds for any $\Delta$. This can only be true for $A \equiv 0$ and $R \equiv 0$. This degeneracy in itself does not violate (ii), since the probability constraints require $W = M$, as established earlier.

## H  EqIM(1)$\wedge$EqIM(2+)$\not\to$EqIM(3) for full low rank $M$?

The following numerical approach may lead to counter-examples with full support to (v) without any divisions by 0 ($M^+_{ss'} > 0$ and $W^+_{ss'} > 0 \,\forall ss'$). We now consider full $M^a$ but of rank $r < d$. The most interesting case is where all $\dot{M}^a$ span the same row-space, i.e. $M^a = L^a \cdot R$, where $L^a$ are $d \times r$ matrices and $R$ is a $r \times d$ matrix. Recall $A' := \binom{A^\cdot}{C^\cdot}$ with $A^a$ and $C^a$ defined in (17) and (18). Empirically, for $k=2$, the rank of $A'$ typically is $\min\{d^2, (3r-1)d - r(r-1)\}$, never more, and only in degenerate cases less. Hence for $r=2$, $A'$ is singular for $d \geq 5$. Hence for $d \geq 5$, there exist $\Delta \neq 0$ with $A'\Delta = 0$,

For $\Delta_0 := \Delta = \Theta(\varepsilon)$, this is an approximate $\Theta(\varepsilon^2)$ solution of $A'\Delta = R'(\Delta)$. By iterating $\Delta \leftarrow \Delta_0 + A'^+ R'(\Delta)$, where $A'^+$ is the pseudo-inverse of $A'$, we get an $\Theta(\varepsilon^i)$-approximation after $i-2$ iterations. This should rapidly converge to an "exact" non-zero(!) solution $A'\Delta = R'(\Delta)$. This would show that (ii) can fail for full $M$. Generically, this solution also violates EqIM(3), i.e. also (vi) can fail. By this we mean, for randomly sampled $L^a$ and $R$ (for $a = r = 2$ and $d \geq 5$) and performing the procedure above, EqIM(3) does not hold. There is a caveat with this argument, namely if $R'$ is not in the range of $A'$, then this construction fails.

## I  EqIM(1) does not imply EqIM(2) ($\odot$-version)

We have already given a simple example that violates (v) in Section 3, but the example and methodology provided here generalizes to (vi) and even larger $i$. We consider deterministic reversible

666 forward dynamics for any policy $\pi(a|s) > 0$ $\forall as$. For simplicity we assume $k=2$ and uniform policy
667 $\pi(a|s) = \frac{1}{2}$. We defer a discussion of $0/0$ to the end of the next Appendix.

668 We consider $M^a$ and $W^a$ that permute states. That is, $M^{\cdot}_{ss'} := [\![ s' = \pi^{\cdot}(s) ]\!]$ and $W^{\cdot}_{ss'} := [\![ s' = \sigma^{\cdot}(s) ]\!]$
669 for some permutations $\pi^{\cdot}, \sigma^{\cdot} : \{1,...,d\} \to \{1,...,d\}$. Strictly speaking, we should multiply this
670 by $\pi(a|s) = \frac{1}{k}$, but this global factor plays no role here, so will be dropped everywhere. Matrix
671 multiplication corresponds to permutation composition: $[M^{\cdot}W^{\cdot}]_{ss''} = [\![ s'' = \sigma^{\cdot}(\pi^{\cdot}(s)) ]\!]$. We denote
672 example permutation (matrices) by $[\pi] = [\pi(1)...\pi(d)]$.

673 We now construct a counter-example for (v): For $d=4$, let $M^0 = W^0 = \text{Id} = [1234]$ be the identity
674 matrix/permutation. Let $W^1 = [2341]$ be the cyclic permutation $1 \to 2 \to 3 \to 4 \to 1$, and $M^1 = [2143]$
675 the cycle pair $1 \leftrightarrow 2$ and $3 \leftrightarrow 4$. We know from (15) that EqIM(1) holds *iff* $W^a \oslash M^a$ is independent
676 $a \ (=J)$ *iff* $W^a \oslash M^a = W^b \oslash M^b$ $\forall a,b \in \{0,1\}$ *iff* $W^a \odot M^b = M^a \odot W^b$. Case $a=b$ is trivial,
677 so only $W^0 \odot M^1 = M^0 \odot W^1$ needs to be verified. Now $M^{\cdot} \odot W^{\cdot}$ of two permutations matrices
678 is *not* a permutation matrix (unless $M^{\cdot} = W^{\cdot}$). It still a 0-1 matrix with at most one non-zero
679 entry in each row and column. We can generalize the permutation notation to "sub-permutations"
680 by defining $\pi(s) = \emptyset$ if row $s$ is empty. For instance $M^1 \odot W^1 = [2\emptyset 4\emptyset]$. EqIM(1) holds, since
681 $W^0 \odot M^1 = [\emptyset\emptyset\emptyset\emptyset] = M^0 \odot W^1$.

682 Similarly EqIM(2a) holds *iff* $W^a W^{a'} \oslash M^a M^{a'}$ is independent $a,a'$ *iff*

$$W^a W^{a'} \odot M^b M^{b'} = M^a M^{a'} \odot W^b W^{b'} \quad \forall a,a',b,b'. \tag{19}$$

683 But for $a=a'=0$ and $b=b'=1$ we have

$$(W^0)^2 \odot (M^1)^2 = [1234] \odot [1234] = [1234] \neq [\emptyset\emptyset\emptyset\emptyset] = [1234] \odot [3412] = (M^0)^2 \odot (W^1)^2$$

684 hence EqIM(1) does not necessarily imply EqIM(2). The advantage of formulation (19) over (8) is
685 that matrix sums $M^+$ and $W^+$ are more complicated objects than the sub-permutation matrices (19).
686 Like random matrices, permutation matrices, have full rank, but unlike random matrices they can
687 violate (ii), (iv), and (vi).

# J  EqIM($1a$) $\wedge$ ... $\wedge$ EqIM($ia$) do not imply EqIM($i+1$) ($\odot$-version)

689 Counting variables and equations made the possibility of violating (v) for $k < d$ plausible (cf. positive
690 result for $k \geq d$). A similar counting argument indicates that (vi) and higher $i$ analogues might actually
691 hold. Unfortunately this is not the case. I.e. even providing inverse models for all action sequences up
692 to length $i$ is not sufficient to always uniquely determine the probability of longer action sequences.
693 This is true even for deterministic reversible forward dynamics for any policy $\pi(a|s) > 0$ $\forall as$. As for
694 $i=1$, we assume $k=2$, $\pi(a|s) = \frac{1}{2}$, gloss over $0/0$, and don't normalize $M$ and $W$.

695 For $i=2$, $M^0 := W^0 := \text{Id} = [123456]$ and $W^1 := [234561] =: \sigma$ ($\sigma$ for 'cycle') and $M^1 := [231564] =: \pi$
696 can be shown to satisfy EqIM(1) and EqIM(2a) but violate EqIM(3). The calculations are not to
697 onerous, but lets consider directly the general $i$ case: Consider even $d =: 2d'$ and identity and cycle
698 (pair)

$$M^0 = W^0 = \text{Id} = [1,2,...,d-1,d],$$
$$W^1 = [2,3,...,d,1], \quad M^1 = [2,3,...,d',1,d'+2,...d-1,d,d'+1]$$

699 EqIM($ia$) holds *iff* $W^a W^{a'}... \oslash M^a M^{a'}... = W^+ W^+... \oslash M^+ M^+...$ is independent $aa'...$ *iff*

$$W^a W^{a'}...W^{a^i} \odot M^b M^{b'}...M^{b^i} = M^a M^{a'}...M^{a^i} \odot W^b W^{b'}...W^{b^i} \quad \forall aa'...a^i, bb'...b^i \tag{20}$$

700 (While this looks like $k^{2i}$ matrix equations, by chaining, checking $k^i$ pairs suffices, which is the
701 same number as in EqIM($ia$)). Now $W^a W^{a'}...W^{a^i}$ consists of only two types of matrices, a
702 cycle for $W^1 = \sigma$ and identity $W^0$. The $W^0 = \text{Id}$ can be eliminated, leading to $(W^1)^{a^+}$, where
703 $a^+ := a + a' + ... + a^i$. Similarly $M^b M^{b'}...M^{b^i} = (M^1)^{b^+}$, etc. Hence we only need to verify

$$(W^1)^{a^+} \odot (M^1)^{b^+} = (M^1)^{a^+} \odot (W^1)^{b^+} \quad \text{for} \quad 0 \leq a^+, b^+ \leq i \tag{21}$$

$$(W^1)^{a^+} = [a^+ +1, a^+ +2, ..., d, 1, 2, ..., a^+], \quad \text{while}$$
$$(M^1)^{b^+} = [b^+ +1, ..., d', 1, ..., b^+, d'+1+b^+, ..., d, d'+1, ..., d'+b^+]$$

hence $(W^1)^{a^+} \odot (M^1)^{b^+} = [\emptyset ... \emptyset] = 0$ for $0 \le a^+ \ne b^+ < d'$. For $a^+ = b^+$ both sides of (21) are equal too. Hence if we choose $d' = i+1$, (21) and hence EqIM(1)...EqIM($ia$) are all satisfied. If we choose $d' = i$, $a^+ = d'$, $b^+ = 0$, (21) reduces to

$$(W^1)^{d'} \odot (M^1)^0 = [d'+1, ..., d, 1, ..., d'] \odot \mathrm{Id} = 0, \quad \text{and}$$
$$(M^1)^{d'} \odot (W^1)^0 = \mathrm{Id} \odot \mathrm{Id} = \mathrm{Id}$$

which are of course not equal. Hence EqIM($i$) fails for $d' = i$. Summing over all $a'...a^{d'}$ and $b'...b^{d'}$, and noting that all other terms are $0$ or cancel, shows that EqIM($i+$) fails too. Together this shows for $d' = i+1$ that EqIM(1)...EqIM($ia$) do not imply any version of EqIM($i+1$).

Despite $M^a$ having full rank, $A$ and $A'$ defined in Appendix G have very low rank, indicating potentially many more consistent $W$.

A downside of this example is that it strictly only applies to the $\odot$-version (20). Many entries of $M^+$ and $W^+$ and powers thereof are $0$, so (8) contains many divisions by zero. We were not able to extend this example by mixing in e.g. a uniform matrix as done in the first counter-example to (v).

Many real-world MDPs are sparse. Only a subset $G \subseteq S \times S$ of transitions $s \to s'$ is possible. For $(s, s') \notin G$, $p(s'|sa) = 0 \, \forall a$, or formally $\dot{M}^a_{ss'} = M^+_{ss'} = 0$. In this case, no action causes $s \to s'$ and $p(a|ss') = M^a_{ss'} / M^+_{ss'}$ being undefined is actually appropriate. So we could restrict $(s, s')$ to $G$ (and analogously $(s, ..., s^i)$ and $(ss^i)$ by chaining $G$) in the conditions and conclusions of the various conjectures. It is then also natural to restrict the model class to $\mathcal{M} := \{M^{\cdot} : M^+_{ss'} > 0 \Leftrightarrow (s, s') \in G\}$. For unknown $G$, the condition $M, W \in \mathcal{M}$ then becomes $M^+_{ss'} > 0 \Leftrightarrow W^+_{ss'} > 0$. Unfortunately the above counter-example does not even satisfy this weaker condition, but the more complicated example of Appendix K does. See Appendix Q for how to treat 0/0 in practice.

# K Non-Uniqueness of Inverse MDP Models for $i \ge 2$

In Appendices I/J we provided conjectured/unsatisfactory counter-examples to EqIM($1:i$)$\Rightarrow$EqIM($i+1$). Here we provide a fully satisfactory counter-example that avoids the "bad" 0/0.

**EqIM(1) and EqIM(2a) do not imply EqIM(3).** Consider two matrices $\dot{M}^0$ and $\dot{M}^1$ with disjoint support, i.e. $\dot{M}^0 \odot \dot{M}^1 = 0$. In this case $\dot{M}^a \oslash \dot{M}^+ \in \{0, 1, \perp\}^{d \times d}$ is a partial binary matrix with entry undefined ($\perp$) wherever $\dot{M}^+ = 0$ but otherwise $0$ wherever $\dot{M}^a = 0$ and $1$ wherever $\dot{M}^a > 0$. That is, it is insensitive to the actual (non-zero) values of $\dot{M}^a$. A simple such $\dot{M}$ is $\dot{M}^0 = \begin{pmatrix} 1 & 0 \\ 0 & 1 \end{pmatrix}$ and $\dot{M}^1 = \begin{pmatrix} 0 & 1 \\ 1 & 0 \end{pmatrix}$, ignoring normalization. For now we ignore $ss'$ for which $\dot{M}^+_{ss'} = 0$ and return to this issue later.

We consider $M^a$ and $W^a$ that permute states. That is, $M^{\cdot}_{ss'} := [\![s' = \pi^{\cdot}(s)]\!]$ and $W^{\cdot}_{ss'} := [\![s' = \sigma^{\cdot}(s)]\!]$ for some permutations $\pi^{\cdot}, \sigma^{\cdot} : \{1, ..., d\} \to \{1, ..., d\}$. Strictly speaking, we should multiply this by e.g. $\pi(a|s) = \frac{1}{k}$, but this global factor plays no role here, so will be dropped everywhere. Matrix multiplication corresponds to permutation composition: $[M^{\cdot} W^{\cdot}]_{ss''} = [\![s'' = \sigma^{\cdot}(\pi^{\cdot}(s))]\!]$. We denote example permutation (matrices) by $[\pi] = [\pi(1)...\pi(d)]$. Consider now

$$
\begin{aligned}
\dot{M}^0 &:= [456123] =: [\pi_0] \\
\dot{M}^1 &:= [231645] =: [\pi_1]
\end{aligned}
\quad \Longrightarrow \quad
\begin{aligned}
\dot{M}^0 \dot{M}^0 &= [123456] \\
\dot{M}^0 \dot{M}^1 &= [564312] \\
\dot{M}^1 \dot{M}^0 &= [645231] \\
\dot{M}^1 \dot{M}^1 &= [312564]
\end{aligned}
\tag{22}
$$

No column contains the same number twice, hence this not only satisfies $\dot{M}^0 \odot \dot{M}^1 = 0$ but also

$$\dot{M}^a \dot{M}^{a'} \odot \dot{M}^b \dot{M}^{b'} = 0 \quad \text{unless } a = b \text{ and } a' = b' \tag{23}$$

739 That $6\to5\to4\to6$ is in reverse oder to $1\to2\to3\to1$ is crucial for making $\dot{M}^0$ and $\dot{M}^1$ not commute.
740 Note that (23) remains valid if each 1-entry of $\dot{M}^a$ is replaced by a different non-zero scalar, since
741 (23) is purely multiplicative. So if $\dot{W}^a=\dot{M}^a\odot\dot{J}$ for some $J>0$, then $\dot{W}^a\dot{W}^{a'}=\dot{M}^a\dot{M}^{a'}\odot K$ for
742 some $K>0$. Let $\dot{W}^a$ be such a matrix. Then $[\dot{W}^a\dot{W}^{a'}\oslash\dot{W}^+\dot{W}^+]_{\dot{s}\dot{s}''}=1$ if $[\dot{M}^a\dot{M}^{a'}]_{\dot{s}\dot{s}''}>0$ and 0
743 (or undefined) otherwise, i.e. is independent of the choice of $J$. So such $\dot{W}\neq\dot{M}$ satisfies EqIM($2a$).
744 Unfortunately the probability constraints $W^a_{s+}=1$ require $J^a_{ss'}=1$ when $M^+_{ss'}>0$, and hence $W=M$.
745 But the general idea is sound and can be made work as follows:

746 We split one state, e.g. $s=6$ into two states $s=6a$ and $s=6b$. We leave the permutation structure
747 intact, except that all deterministic transitions into $s=6$ are split into stochastic transitions to $s=6a$
748 and $s=6b$, and transitions from $6a$ and $6b$ will be to the same state as from original 6. Condition (23)
749 is still satisfied, so the above argument still goes through, but now we can choose different stochastic
750 transitions to $s=6a$ and $s=6b$ in $W$ and $M$.

751 Finally, we have to show violation of EqIM(3). EqIM($ia$) holds *iff* $W^aW^{a'}...\oslash M^aM^{a'}... = $
752 $W^+W^+...\oslash M^+M^+...$ is independent $aa'...$ *iff*

$$W^aW^{a'}...W^{a^i}\odot M^bM^{b'}...M^{b^i} \;=\; M^aM^{a'}...M^{a^i}\odot W^bW^{b'}...W^{b^i} \quad \forall aa'...a^i,bb'...b^i \qquad (24)$$

753 (While this looks like $k^{2i}$ matrix equations, by chaining, checking $k^i$ pairs suffices, which is the same
754 number of equations as in EqIM($ia$)).

755 It is easier to split *every* state into two states: $s:=(\dot{s},\ddot{s})$ with $\dot{s}\in\{1,...,6\}$ as before and splitter
756 $\ddot{s}\in\{0,1\}$. $M^a_{ss'}:=\dot{M}^a_{\dot{s}\dot{s}'}\ddot{M}^{a\dot{s}}_{\ddot{s}\ddot{s}'}$. Note that $\ddot{M}$ is flexible enough to expand each 1-entry in $\dot{M}^a$ to a
757 different $2\times2$ (stochastic) matrix, while the 0-entries become $\left(\begin{smallmatrix}0&0\\0&0\end{smallmatrix}\right)$. This flexibility is important: $\ddot{M}$
758 independent $a$ or independent $\dot{s}$ would not work. Now let us write out

$$[M^aM^{a'}M^{a''}]_{ss'''} \;=\; \sum_{\dot{s}'\dot{s}''}\dot{M}^a_{\dot{s}\dot{s}'}\dot{M}^{a'}_{\dot{s}'\dot{s}''}\dot{M}^{a''}_{\dot{s}''\dot{s}'''}\sum_{\ddot{s}'\ddot{s}''}\ddot{M}^{a\dot{s}}_{\ddot{s}\ddot{s}'}\ddot{M}^{a'\dot{s}'}_{\ddot{s}'\ddot{s}''}\ddot{M}^{a''\dot{s}''}_{\ddot{s}''\ddot{s}'''} \qquad (25)$$

759 The crucial difference to the $i=2$ case (23) is that now there are difference permutation sequences
760 leading to the same permutation, for instance $\dot{M}^0\dot{M}^0\dot{M}^1=\dot{M}^1=\dot{M}^1\dot{M}^0\dot{M}^0$. Let us choose
761 $aa'a''=001$ and $\dot{s}=1$, then only $\dot{s}'=\pi_0(\dot{s})=4$ and $\dot{s}''=\pi_0(\dot{s}')=1$ contribute to the sum and
762 $\dot{s}'''=\pi_1(\dot{s}'')=2$. For this choice, (25) becomes $1\cdot1\cdot1\cdot[\ddot{M}^{01}\ddot{M}^{04}\ddot{M}^{11}]_{\ddot{s}\ddot{s}'''}$. If we replace $aa'a''$ in
763 (25) by $bb'b''$ and then choose $bb'b''=100$ and again $\dot{s}=1$, then only $\dot{s}'=\pi_1(\dot{s})=2$ and $\dot{s}''=\pi_0(\dot{s}')=5$
764 contribute and $\dot{s}'''=\pi_0(\dot{s}'')=2$. For this choice, (25) becomes $1\cdot1\cdot1\cdot[\ddot{M}^{11}\ddot{M}^{02}\ddot{M}^{05}]_{\ddot{s}\ddot{s}'''}$. We now
765 define $W^a_{ss'}:=\dot{M}^a_{\dot{s}\dot{s}'}\ddot{W}^{a\dot{s}}_{\ddot{s}\ddot{s}'}$. Since $\dot{M}$ remains the same, the same action and state sequences above
766 lead to the same result for $W$, just with $\ddot{M}$ replaced by $\ddot{W}$. If we plug the four expressions into (24)
767 (for $i=3$) we get

$$\ddot{W}^{01}\ddot{W}^{04}\ddot{W}^{11}\odot\ddot{M}^{11}\ddot{M}^{02}\ddot{M}^{05} \;=\; \ddot{M}^{01}\ddot{M}^{04}\ddot{M}^{11}\odot\ddot{W}^{11}\ddot{W}^{02}\ddot{W}^{05}$$

768 Since this expressions involves 10 different $2\times2$ stochastic matrices, there are plenty of choices to
769 make both sides different. If we choose all $2\times2$ matrices to have full support, then by construction,
770 $W$ and $M$ have the same support, hence constitute a proper counter-example to EqIM(3). We now
771 extend this construction to $i>2$.

**EqIM($1a$)$\wedge...\wedge$EqIM($ia$) do not imply EqIM($i+1$).** The construction in the previous para-
773 graph generalizes to $i>2$: We need to find two permutations $\dot{M}^0=\pi_0$ and $\dot{M}^1=\pi_1$ such that for each
774 fixed $j\leq i$ all possible $2^j$ concatenations (products) of these permutation (matrices) differ in the sense
775 that no $s$ is mapped to the same $s^j$ (they have disjoint support). Since all $\dot{M}^a\dot{M}^{a'}...\dot{M}^{a^j}\in\{0,1\}$, we
776 can write this condition compactly as

$$\sum_{aa'...a^j}\dot{M}^a\dot{M}^{a'}...\dot{M}^{a^j} \;\in\; \{0,1\}^{d\times d}$$

777 By factoring the sum, this is equivalent to $(\dot{M}^+)^j\in\{0,1\}^{d\times d}$. Note that $[(\dot{M}^+)^j]_{ss^i}$ counts the
778 number of action sequences $aa'...a^j$ of length $j$ that lead from $s$ to $s^i$. For $j=i+1$, we want this
779 condition to be violated. So in order to disprove the implication we need to find two permutations
780 $M^0$ and $M^1$ such that

$$(\dot{M}^+)^j\in\{0,1\}^{d\times d} \quad \forall j\leq i \quad \text{but} \quad (\dot{M}^+)^{i+1}\notin\{0,1\}^{d\times d} \qquad (26)$$

The rest of the argument is the same as for the $i=2$ case above: creating two versions $M^a$ and $W^a$ of $\dot{M}^a$ by spitting one or all states into two, and replacing the 1s by $2\times 2$ different stochastic matrices. As for the choice of $\dot{M}^a$, for $i=3$ we can choose 3-cycle and 5-cycle

$$
\begin{aligned}
\dot{M}^0 &= [6,7,8,9,10,11,12,13,14,15,1,2,3,4,5] \\
&= (1,6,11)(2,7,12)(3,8,13)(4,9,14)(5,10,15) \\
\dot{M}^1 &= [2,3,4,5,1,8,9,10,6,7,14,15,11,12,13] \\
&= (1,2,3,4,5)(6,8,10,7,9)(11,14,12,15,13)
\end{aligned}
\tag{27}
$$

where we also provide the more conventional cycle notation in round brackets. Crucially the 5-cycles have been chosen to not commute with the 3-cycles ($M^0 M^1 \neq M^1 M^0$). Conditions (26) can easily be verified numerically. For higher $i$ we need $p$ cycles and $q$ cycles, where $p$ and $q$ are relative prime and sufficiently large. We need at least $d = p \cdot q \geq 2^i$, otherwise $\dot{M}^+ \notin \{0,1\}^{d\times d}$ by a simple pigeon-hole argument. To prove $\text{EqIM}(1a) \wedge ... \wedge \text{EqIM}(ia) \not\Rightarrow \text{EqIM}(i+1)$ in general for arbitrarily large $i$, we need to invoke some group theory. All-together we have shown that

**Proposition 4 ((i)-(vi) can fail)** *EqIM*$(1a) \wedge ... \wedge$ *EqIM*$(ia)$ *do not necessarily imply EqIM*$(i+1)$ *for any $i$. This in turn implies that (i)-(vi) each can fail for some $M^\cdot$.*

## L  Computational Complexity

Maybe even just characterizing all $M$ for which EqIM(1) and EqIM(2) uniquely determine $W$ is hopeless, not to speak of finding some or all $W$ in case not. More formally, we can ask the question of whether there exists an efficient algorithm that can decide whether EqIM($i$) has a unique solution. We provide some weak preliminary evidence, why this problem may be NP-hard. Appendix O contains fully self-contained a few versions of this open problem in their simplest instantiation and most elegant form.

**Decidability and computability.**  EqIM(2) converted to (24) and (7), or (28) or (29) below form a System of Quadratic Equations (SQE). The constraint $W \neq M$ can also be expressed as a quadratic equation (see below). As such, the existence and uniqueness of solutions is formally decidable by computing a Gröbner basis [Stu02], and (some) solutions can be found by cylindrical algebraic decomposition in (double) exponential time. $\varepsilon$-approximate solutions can of course be found by exponential brute-force search through all $W$ on a finite $\varepsilon'$-grid, and verified in polynomial time.

**Complexity considerations.**  3SAT is NP complete. A CNF formula in $n$ boolean variables can easily be converted to a System of Quadratic Equations (SQE). Therefore SQE is also NP hard. EqIM(2+) explicitly written in quadratic form

$$
M^a M^+ \odot (W^+)^2 - W^a W^+ \odot (M^+)^2 = 0
\tag{28}
$$

constitutes an SQE in $W$ given $M$, also if we include linear EqIM(1) and probability constraints (7). Non-negativity of $W$ can be enforced with (slack) variables $(Y^a_{ss'})^2 = W^a_{ss'}$. (Similarly (17) plus constraints (16) constitute an SQE in $\Delta$.) To reduce the uniqueness question to a solvability problem we need to avoid the trivial solution $W \equiv M$, e.g. by introducing further (slack) variables $t \in \mathbb{R}$ and $\Gamma^a_{ss'} := (W^a_{ss'} - M^a_{ss'})^2$ and constraint $t \cdot \Gamma^+_{++} = 1$. Due to the minus sign in (28), this can*not* be converted to a convex (optimization) problem. The choice of $M$ gives significant freedom in creating SQE problems, even if only considering permutation matrices $M^a \in \{0,1\}^{d\times d}$. If one could show that every SQE can be represented as (28) [plus $W \neq M$ constraint] for a suitable choice of $M$, this would imply that proving the existence of $W \neq M$ satisfying (28) is NP hard. This in turn would imply that computing (any) $p(a|ss''')$ from $p(a|ss')$ and $p(a|ss'')$ is NP hard. On the other hand, matrix multiplication $W^a W^b$ is a very specific quadratic form, which may not be flexible enough to incorporate every SQE within (28).

We could not find any work on NP-hardness of Systems of Polynomial Matrix Equations (SPME). There is work on the NP-hardness of tensor problems [HL13], but this refers to the design tensors, e.g. $\sum_{jk} A_i^{jk} x_j x_k + \sum_j B_i^j x_j + C_i = 0 \; \forall i$, but the unknowns are always treated as scalars or vectors. Of course $[X \cdot Y]_{ik} = \sum_{abcd} A^{abcd}_{ik} X_{ab} Y_{cd}$, but $A^{abcd}_{ik} = \delta_{ai} \delta_{dk} \delta_{bc}$ is a very special fixed tensor (actually of low tensor rank $d$) with no flexibility of encoding NP-hard problems therein.

825 That inference in Bayesian networks is NP-complete [KF09] does not help us either for two reasons:
826 First, in our problem the probability distribution over states and actions is only partially given. More
827 importantly, our network for $i=2$ has only 5 nodes $(s,a,s',a',s'')$, while the NP-hardness proofs we
828 are aware of require large networks. Even for fixed $i>2$, it is not obvious how to encode NP-hard
829 problems into EqIM(i), due to the severe structural constraints in EqIM(i) compared to a general
830 network with $2i+3$ nodes. It is not clear how to exploit the fact that our (few) state nodes are large.

831 SQE are polynomially equivalent to Systems of Quadratic Matrix Equations (SQME), which may be
832 the reason complexity theorists have ignored the latter. We suspect but do not know whether SQME
833 of *bounded* structural complexity (only the definitions of the constant matrices scale with $d \times d$) is
834 NP-hard (Open Problem 7). If we allow sparse encoding of SQE variables in $W$, i.e. we allow one
835 equation involving $\odot$ of the form $B \odot W = 0$ with boolean matrix $B$, then bounded SQME becomes
836 NP-hard. See Appendix M for details.

837 Below we directly reduce 1in3SAT to a Bounded-SQME with $\odot$ that resembles our problem as close
838 as we were able to make it.

839 **An NP-hard matrix problem.**   From EqIM(1) we know that $W^a = B^a \odot W^+$. Plugging this into
840 EqIM($2a$) gives

$$B^{aa'} \odot (W^+ \cdot W^+) = (B^a \odot W^+)(B^{a'} \odot W^+) \quad \text{with constraints} \quad [B^a \odot W^+]_{s+} = \pi(a|s) \quad (29)$$

841 This set of equations is purely in terms of what is given ($B^a$ and $B^{aa'}$) and only involves unknowns
842 $W^+$ without reference to $W^a$. See Appendix N for some further simplification and discussion. We
843 will show:

844 **Proposition 5 (An NP-hard matrix problem)** *Given $A,B,C,\Pi$, deciding whether the following*
845 *quadratic matrix problem has a solution in $W$ is NP-hard:*

$$A \odot (W \cdot W) = (C \odot W)(C \odot W), \quad [B \odot W]_{s+} = 1, \quad \Pi \cdot W = W \quad (30)$$

846 This has some resemblance to (29). Since the boundary between P and NP is very fractal/subtle,
847 this in-itself may not imply much, but is more meant as a demonstration of how one may approach
848 proving NP-hardness of (29).

849 **Proof.** We reduce 1in3SAT, which is an NP-complete variant of 3SAT, where each clause must have
850 exactly one satisfying assignment, to (30). A 3CNF($n,m,g$) formula is a boolean conjunction of $m$
851 clauses in $n$ variables, where each clause $c_i = \ell_{i1} \dot\vee \ell_{i2} \dot\vee \ell_{i3}$ for $i \in \{1:m\}$ is a 1-in-3 disjunction of 3
852 literals, and each literal is $\ell_{ia} = x_j$ or it's complement $\ell_{ia} = \neg x_j \equiv \bar{x}_j$, where $j = g(i,a)$ is the variable
853 index of clause $i$ in position $a$.

854 We arithmetize the 3CNF expression in the standard way by replacing True$\rightsquigarrow 1$, False$\rightsquigarrow 0$, and $\dot\vee \rightsquigarrow +$,
855 i.e. we ask whether the system of linear equations $\ell_{i1} + \ell_{i2} + \ell_{i3} = 1 \ \forall i$ has a solution in $x_j \in \{0,1\}$.
856 We need to encode the $x$'s into $W$ somehow: We aim at the following embedding:

$$W = \begin{pmatrix} x_1 & \bar{x}_1 & ... & x_n & \bar{x}_n & y_0 & ... & y_k \\ \vdots & \vdots & \ddots & \vdots & \vdots & \vdots & \ddots & \vdots \\ x_1 & \bar{x}_1 & ... & x_n & \bar{x}_n & y_0 & ... & y_k \end{pmatrix}$$

857 The $y$ are $k+1 := \max\{1,m-n+2\}$ extra dummy variables to make the matrix a square $d \times d$ matrix
858 with $d := \max\{m+n+2, 2n+1\}$.

859 Choosing a cyclic permutation matrix $\Pi = [234...d1]$ ensures that all rows of $W$ are indeed the same
860 via $\Pi \cdot W = W$. The standard way of achieving $x_j, y_j \in \{0,1\}$ is via $x_j^2 = x_j$ and $y_j^2 = y_j$. This can be
861 achieved via $(\text{Id} \odot W)^2 = \text{Id} \odot W$, were $\text{Id}_{ss'} = \delta_{ss'}$ is the identity matrix.

862 We use $[B \odot W]_{s+} = 1$ to ensure $\bar{x}_j = 1 - x_j$, $y_0 = 1$, and $y_1 = ... = y_k = 0$ and $\ell_{i1} + \ell_{i2} + \ell_{i3} = 1$ by
863 setting $B_{s,2s-1} = B_{s,2s} = 1$ for $s \in \{1:n\}$, and $B_{i+n,2j-1} = 1$ if $\ell_{ia} = x_j$ and $B_{i+n,2j} = 1$ if $\ell_{ia} = \neg x_j$
864 for $i \in \{1:m\}$ and $a \in \{1,2,3\}$, and $B_{d-1,2n+1} = ... = B_{d-1,2n+m} = 1$, and $B_{d,2n+1} = 1$, and $B_{ss'} = 0$
865 for all other $ss'$. This also ensures that all rows of $W$ sum to $n+1$, hence $W \cdot W = (n+1)W$, so
866 $x_j \in \{0,1\}$ can be achieved via $C = \text{Id}$ and $A = \frac{1}{n+1}\text{Id}$ in $A \odot (W \cdot W) = (C \odot W)(C \odot W)$.

867 The construction implies that the 3CNF($n,m,g$) formula is satisfiable *iff* (30) has a solution in $W$
868 with the $A,B,C,\Pi$ as constructed above. This shows NP-hardness of deciding whether (30) has a

solution. A solution can trivially be verified (in the rationals or to $\varepsilon$-precision over the reals) in time $O(d^3)$, hence the problem is in NP, hence NP-complete.

## M  Systems of Quadratic Matrix Equations

A System of Polynomial Equations (SPE) is a set of multivariate polynomial equations $\text{Poly}_j(x,y,z,...)=0$ over $\mathbb{R}$ in $n$ variables $x,y,z,u,v,w,... \in \mathbb{R}$ for $j \in \{1:m\}$. This class is NP-hard (via a simple reduction from 1in3SAT, see Section L). We can recursively replace each product $xy$ (sum $bu+cv$) in the polynomials by a new variable $z$ ($w$) and add "polynomial" equation $z=xy$ ($w=bu+cv$). This results in SPEs consisting of only linear equations with a single $+$ ($bu+cv=w$) and quadratic equations without any $+$ ($xy=z$), which are still (even with all $a=b=1$ and $x=y=z$) NP-hard. We call them Simple Systems of Quadratic Equations (Simple SQE). For the reduction process to actually work we need one further dummy variable and equation $q=1$ (to reduce $bu+c=w$). Alternatively, with some extra work, we can reduce any SPE into a Simple SQE asking for a *non-zero* solution. We will pursue the latter, since this is closer to our interest (SQE (17) with solution $\Delta \not\equiv 0$). We can even merge the linear and quadratic equations into a single form $xy=bu+cv$ by choosing $b=1$ and $c=0$ (replacing $xy$ by $w$ and adding $xy=0{\cdot}u+1{\cdot}w$).

We define a System of Polynomial/Quadratic Matrix Equations (SPME/SQME) as a set of $m$ multivariate (quadratic) polynomials $\text{Poly}_j(\Delta,\Gamma,...|A,B,C,...)=0$ in the (unknown) matrix variables $\Delta,\Gamma,...$ *and* the (given) matrix constants ("coefficients") $A,B,C,...$. Alternatively, $\text{Poly}_j$ might be viewed as generalized polynomials over a *non*-commutative matrix ring in the unknowns only. In any case, note that

$$A{\cdot}\Delta{\cdot}A'{\cdot}\Delta{\cdot}A''+B{\cdot}\Delta{\cdot}B'+C \;\neq\; (A{\cdot}A'{\cdot}A''){\cdot}\Delta^2+(B{\cdot}B'){\cdot}\Delta+C$$

By writing out all matrix operations in terms of their scalar operations, SPME is of course a sub-class of SPE. SPE is also a sub-class of SPME (choose all matrices to be $1{\times}1$ matrices), which implies SPME is NP-hard. But we are interested in NP-hard *small* subclasses of SPME, so will construct a more economical embedding: Assume we have a Simple SQE with $n$ variables $x,y,z,u,v,....$. We place them into $d{\times}d$ matrix $\Delta$ ($d \geq \sqrt{n}$) introducing dummy variables for the remaining entries. We can extract variable $w=\Delta_{ss'}$ via $w=\text{e}^{s\top}{\cdot}\Delta{\cdot}\text{e}^{s'}$, where $\text{e}^s$ is basis vector ($d{\times}1$-matrix) $(\text{e}^s)_{s'1}=\delta_{ss'}$. If we replace all variables in the Simple SQE expressions $xy=au+bv$ by such expressions, we get a Simple SQME with $\text{Poly}_j$ equations of the form (dropping $\cdot$ as usual)

$$a^j \Delta A'^j \Delta a''^j \;=\; b^j \Delta b'^j + c^j \Delta c'^j \quad \forall j \tag{31}$$

While these are scalar equations, since the outer matrices are $1{\times}d$ on the left and $d{\times}1$ on the right, technically they are matrix equations. We could pad all involved matrices, including the outer ones, with zeros to square $\mathbb{R}^{d \times d}$ matrices of the same size (for sufficiently large $d$, and only polynomial overhead).

We can reduce (31) to just one equation at the cost of making the equations more complicated as follows: Write each equation $\text{Poly}_j=0$ in the form $\text{e}^s{\cdot}\text{Poly}_j{\cdot}\text{e}^{s'\top}=0$, with a different $(s,s')$-pair for each $j$. These are now "proper" matrix equations, but with all entries identically 0 except entry $(s,s')$ being $\text{Poly}_j$. This allows us to sum all equations without conflating them into one (complex) matrix equations

$$\sum_j A^j \Delta A'^j \Delta A''^j \;=\; \sum_j B^j \Delta B'^j + C^j \Delta C'^j \tag{32}$$

Another way to combine (31) into one equation is by putting all $M^j$ for all $j$ into one block-diagonal matrix $\tilde{M}:=\text{Diag}(M^1,...,M^m)$ for $M \in \{a,A',a'',b,b',c,c',\Delta\}$. For $\tilde{\Delta}$ we need to ensure that indeed all blocks $\Delta^j=\Delta$ are equal. This can be done via $\tilde{\Pi}^\top \tilde{\Delta} \tilde{\Pi}=\tilde{\Delta}$ for some cyclic block permutation $\tilde{\Pi}$. We further need to ensure that the off-diagonal blocks of $\tilde{\Delta}$ are zero. We can zero each block with one equation, but it seems impossible to zero all with a bounded number of Simple QMEs. We can modify the decision problem to decide whether specific sparse solutions $\tilde{\Delta}$ exist. Formally, we can introduce element-wise multiplication $\odot$ and allow one equation of the form $\tilde{B} \odot \tilde{\Delta}=0$ with $\tilde{B}$ being $0/1$ on the on/off-diagonal blocks. This leads to a Simple SQME with $\odot$ in 3 equations (dropping the $\sim$)

$$A\Delta A' \Delta A'' \;=\; B\Delta B' + C\Delta C', \quad \Pi^\top \Delta \Pi = \Delta, \quad B \odot \Delta = 0 \tag{33}$$

**Proposition 6 (NP-hardness of Simple SQME)** *Systems of Polynomial Equations (SPE) can be polynomially reduced to Simple Systems of Quadratic Matrix Equations (Simple SQME)* (31). *The number of equations can be reduced to 1 at the expense of making the equations complex* (32), *or to 2 by asking for sparse solutions or by enforcing sparsity via $B \odot \Delta = 0$* (33). *Since SPE are NP-hard, deciding the existence of non-zero solutions for all three SQME versions is also NP-hard.*

An NP-hardness proof for a Simple SQME with $\odot$ with 3 equations via reduction from 1in3SAT that looks much closer to the desired form (29) or (34) is given in Section L. By a similar reduction, encoding all $n$ variables and their complement in the diagonal of $\Delta = \mathrm{Diag}(x, \bar{x}, y, \bar{y}, ...,)$, one can also show that solvability of

$$\Delta^2 = \Delta, \quad A \Delta 1 = 1, \quad \mathrm{Id} \odot \Delta = \Delta, \quad \text{with} \quad A \in \{0,1\}^{m \times 2n}$$

is NP-complete (1 is the all-1 vector, sparse $A$ with 2 or 3 ones in each row suffice), but not all SPE can be reduced to this form.

**Open Problem 7 (Are Bounded SPME NP-hard?)** *Are Systems of Polynomial Matrix Equations (without $\odot$) of bounded structural complexity NP-hard? Bounded means, only the definitions of the constant matrices scale with $d \times d$, but the polynomial degrees, number of equations, and number of matrix operations are bounded.*

# N   Compact Representation of EqIM(2+)

If only $B^{a+}$ (EqIM(2+)) is given, we can sum (29) over $a'$. If we further assume $a = 2$ and define $B = B^0$ and $A = B^{0+}$ and $W = W^+$ and exploit $B^+ = B^{++} = 1$, this reduces to the elegant quadratic matrix equation

$$A \odot (W \cdot W) = (B \odot W) \cdot W \tag{34}$$

with constraints as in (29), or even simpler $W_{s+} = 1$ if $\pi$ is unknown. This is the most pure formulation of the problem we are trying but are unable to solve we could come up with. For $A$ and $B$ defined via $M$, we know that (34) has a solution (namely $W = M^+$).

We neither know whether there exists an efficient algorithm to find *some* solution (34), nor to find *the* solution in case it is unique, nor to decide whether there exist solutions in case $A$ and $B$ are chosen arbitrarily.

The condition $W_{s+} = 1$ can be relaxed to $W_{s+} > 0$. If $W_{ss'}$ is a solution of (34), then also $v_s^{-1} W_{ss'} v_{s'}$ for any $v. > 0$ (most easily checked via (11)). Every non-negative matrix has a real non-negative Eigenvector $v$, and $W_{s+} > 0$ implies $v_s > 0$ and Eigenvalue $\lambda > 0$, hence for $W_{ss'}^{\mathrm{norm}} := (\lambda v_s)^{-1} W_{ss'} v_{s'}$, we have $W_{s+}^{\mathrm{norm}} = 1$.

$B^a \geq 0$ and $B^+ = 1$ iff $B \in [0;1]$ (and $B^1 = 1 - B$). $B^{a+} \geq 0$ and $B^{++} = 1$ iff $A \in [0;1]$ (and $B^{1+} = 1 - A$). But we can scale back any $A$ and $B$ by the same $0 < \lambda < 1$ to satisfy these without changing (34), i.e. these extra conditions ($A$ and $B$ bounded by 1) do not make the problem any simpler.

# O   Open Problem

We present the most important open problem(s) in their simplest instantiation and most elegant form, fully self-contained here: Consider matrices $A, B, W \in [0;1]^{d \times d}$ with $d \in \mathbb{N}$, tied by the quadratic matrix equation

$$A \odot (W \cdot W) = (B \odot W) \cdot W \quad \text{and} \quad W_{s+} = 1 \,\forall s \tag{35}$$

where $\odot$ is element-wise (Hadamard) multiplication and $\cdot$ is standard matrix multiplication. The open problems are as follows: Given $A$ and $B$, are there efficient algorithms which

    (a) decide whether there exists a $W$ satisfying (35)?
    (b) decide whether the solution is unique, assuming (35) has a solution?
    (c) compute *a* solution, assuming (35) has a solution?
    (d) compute *the* solution, assuming (35) has a unique solution?

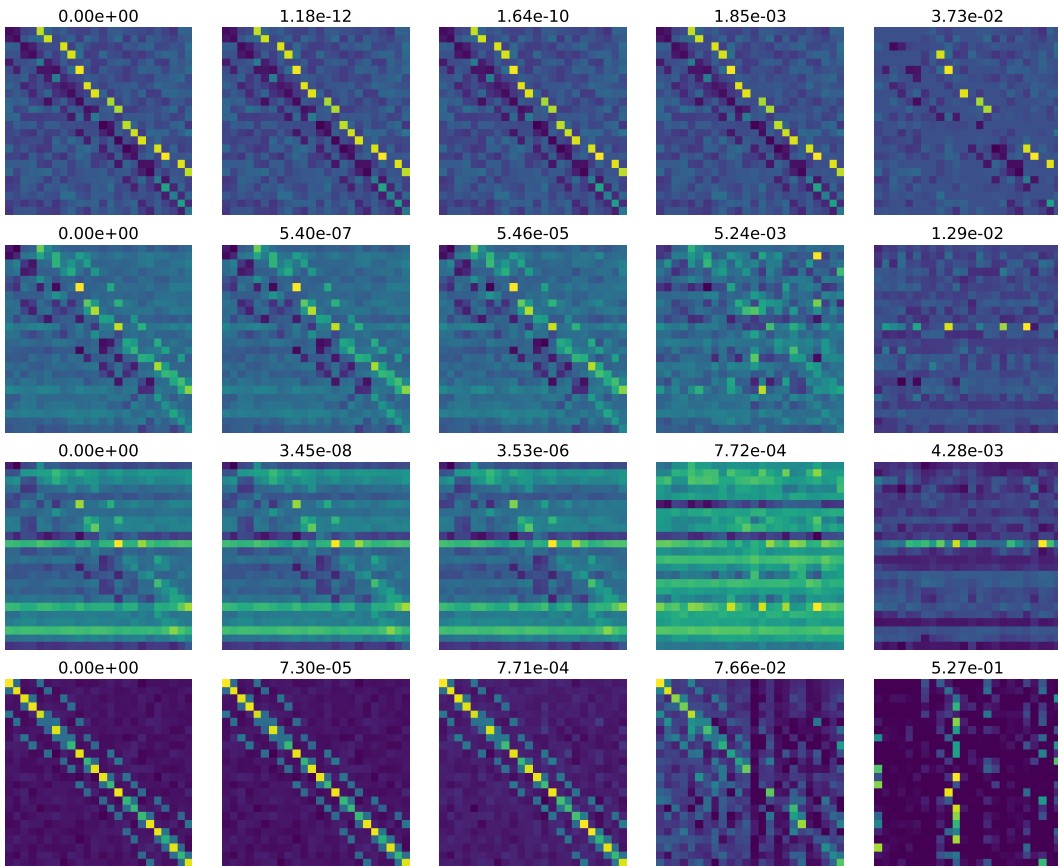

Figure 4: Reconstructing inverse and forward models from inverse models with noise injected. Rows, from top to bottom, show reconstructions of $B^a, B^{a+}, B^{a++}$, and $M^a$. Noise increases exponentially across columns, from left to right, $[0, 10^{-6}, 10^{-5}, 10^{-4}, 10^{-3}]$. The subplot titles show the average KL divergence of the recovered distribution from the ground truth.

Computing a real number means, given any $\varepsilon > 0$, computing an $\varepsilon$-approximation. Efficient means running time is polynomial in $d$, ideally with a degree independent of $1/\varepsilon$. General systems of quadratic equations are known to be NP-hard, but we do not know the complexity of this particular matrix sub-class.

The upper bounds $A, B, W \leq 1$ can always be satisfied by scaling, hence are irrelevant. $W_{s+} = 1$ can be relaxed to $W_{s+} > 0$ except in the uniqueness questions. If helpful: One may assume $A, B, W$ strictly positive. Also, any finite ($d$-independent) number of equations of the form $A' \odot (W \cdot W) = (B' \odot W) \cdot W$ with other *general* matrices $A', B' \in [0;1]^{d \times d}$ may be added, which further constrain the solution space.

# P  Experimental Details

Here we provide further experiments supporting and illustrating the theory. In Appendix Q we show how we numerically dealt with $B = 0/0 = \perp$. Appendix R derives the formulas for the plotted solution dimensions.

**Experiments illustrating robustness to noise.** As mentioned in the main text, rather than committing to a specific learning algorithm, we instead directly inject noise into the true inverse distributions. This is done by adding $\varepsilon \times 10^c$ to the true distribution and renormalizing $B$, where $\varepsilon$ is drawn from the unit uniform distribution: $\varepsilon \sim \mathcal{U}[0,1]$. In Figure 2, this noise is evaluated across several orders of magnitude ($c$ varied $-7$ to $0$).

The main text also mentions that the effect of this noise is substantially diminished as the horizon of the inverse model is increased (from $B1 := B^a$ to $B3 := B^{a++}$). Figure 4 buttresses this interpretation

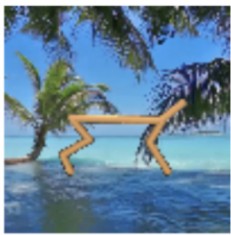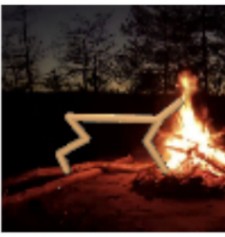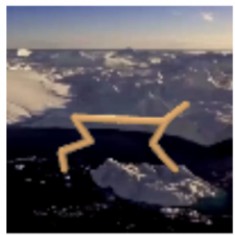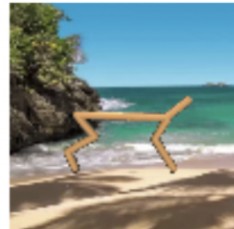

Figure 5: Reproduced from [LFLDP21], this 'half-cheetah' environment has been augmented with videos of complex scenes. This highlights how non-controllable aspects of the environment can be made more complex without changing the underlying control problem. The fact that such environments are of interest motivates our focus on the Tensor-product special case.

by showing that the recovered $B^{a++}$ is qualitatively similar to the ground truth even with substantial noise.

**Experiments on the Tensor-product special case.** As mentioned in the main text, if $M$ factors into two processes $\dot{M}^a \otimes \ddot{M}$, where $\ddot{M}$ is action-independent, then only the complexity of the action-dependent process $\dot{M}^a$ matters for all of our questions.

This particular special case is important because of its frequency in applied work. Many environments have most of their complexity in sub-spaces that the agent has no control over. This is illustrated by Figure 5, reproduced from [LFLDP21], wherein naturalistic videos are superimposed on relatively simple continuous control environments. Clearly, the background dynamics can be arbitrarily complex without impacting the underlying control problem.

We can construct small environments of this form via a simple procedure. We construct $\dot{M}^a$ with $\dot{d}$ states and $k$ actions by sampling each element of the appropriately sized matrices from $\mathcal{U}[0,1]$ and then normalizing. $\ddot{M}$ has $\ddot{d}=2$ states that transition uniformly regardless of the action. For the results shown in Figure 3, $k=5$ as in the main text, and $d=2\dot{d}$ is varied from 2 to 32.

Note that in Figure 3, the solution dimension is non-zero even when $\dot{d} \leq k < d = 2\dot{d}$ (here $k=5$, hence for $d=6|8|10$) despite there necessarily being a unique solution as per Section 4. This is due to the fact that the algorithm does not exploit knowledge of the fact that $M$ is a tensor product, resulting in the solution dimension being correct for the more general case where $W$ is not confined to being tensor product.

# Q How to Deal with 0/0

If for some pair of states $(s,s')$, no action $a$ of positive $\pi$-probability leads from state $s$ to $s'$, i.e. if $M_{ss'}^+ = 0$, then $B_{ss'}^+$ and $B_{ss'}^a, \forall a$ are $0/0 = \bot =$ undefined. To also handle $B_{ss'}^{\cdot} = \bot$, we need to adapt the linear algorithm in Section 4. We provide 2 different ways of doing so, with a couple of variations, all leading to the same correct result.

We have to restrict the sum in $\sum_{s'} B_{ss'}^a J_{ss'} = \pi(a|s)$ to those $s'$ for which $B_{ss'}^a$ is defined. We then solve for $J_{ss'}$, again for $s'$ for which $B_{ss'}^a$ is defined, and set $J_{ss'} = 0$ for those $s'$ for which $B_{ss'}^a = \bot$. Technically this can be achieved by removing the $s'$ columns from matrix $B_{s\cdot}^{\cdot}$ and $J_{\cdot\cdot}$ for which $B_{ss'}^{\cdot} = \bot$, solve the reduced linear equation system, and finally reinsert $J_{ss'} = 0$ for the removed $s'$. Simpler is to replace $B_{ss'}^a = \bot$ by $B_{ss'}^a = 0$, solve the equation for $J$, and then set $J_{ss'} = 0$ for the $s'$ for which the original $B_{ss'}^a$ was $\bot$. Some solvers automatically result in $J_{ss'} = 0$, since this is the minimum norm solution, but it is better not to reply on this. Instead of setting $J_{ss'} = 0$ after solving the linear system, one could augment $B_{s\cdot}^{\cdot}$ with extra rows that enforce $J_{ss'} = 0$.

Alternatively, we could replace $B_{ss'}^{\cdot} = \bot$ by a random vector which sums to 1, e.g. $B_{ss'}^a = r_a/r_+$, where $r_a = -\log u_a$ with $u_a \sim \text{Uniform}[0;1]$. Provided that the solution is unique, this also leads to the correct solution (almost surely), and in this way $J_{ss'} = 0$ automatically. If the solution is not unique, $W^{\cdot}$ will still satisfy $B^a = W^a \oslash W^+$ when for $B_{ss'}^a \neq \bot$, but $W_{ss'}$ may not be 0.

The adaptation of the Linear Relaxation Algorithm in Section 5 follows the same pattern: $A_{ss^i s^j}^{\cdot\cdot} = \bot$ in (12), whenever one of the three involved $B$'s is undefined. For such $ss^i s^j$, we need to ensure that $\hat{U}_{ss^i s^j} = 0$, which can be done with any of the variations described above. Once we have $\hat{U}_{ss^i s^j}$, we set $C_{ss^i s^j}^{a^i} = 0$ if $B_{s^i s^j}^{a^i} = \bot$. No further intervention is needed, since $\hat{U}_{ss^i s^j} = 0$ already.

## R  Solution Dimensions of $W$ and $B^{aa'}$.

In Section 4 we presented an algorithm for inferring $W$ and $B^{aa'}$ from $B^a$. Even if $M$ cannot uniquely be reconstructed ¬(i), $B^{aa'}$ may still be unique (iii). More generally, the solutions $J$ and $W^a$ form linear spaces of dimension $d_J = d_W \leq d(d-1)$ ($d_J \geq d_W$ since $W^{\cdot}$ is a linear function of $J$ and $d_J \leq d_W$ since $W^+ = J$). $B^{aa'}$ is a (non-linear, polynomial) variety of dimension $d_B \leq d_W$ at regular points (it is a smooth function of $W$).

**Parameterizing the solutions for $J$ and $W$ and $B$.** We can determine the solution dimensions $d_J$, $d_W$, and $d_B$ as follows: Let $\Gamma_{ss'}$ be a solution of $[B^a \odot \Gamma]_{s+} = 0$. If $J_{ss'}$ is a solution of $[B^a \odot J]_{s+} = \pi(a|s)$, then so is $\bar{J} := J + \Gamma$, hence $W^a := M^a + \Lambda^a$ is a solution of $B^a = W^a \oslash W^+$ and $W^a_{s+} = \pi(a|s)$, where $M^a := B^a \odot J$ and $\Lambda^a := B^a \odot \Gamma$.

If we plug in $W^a \equiv M^a + \Lambda^a$ into $\bar{B}^{aa'} := W^a W^{a'} \oslash (MW+)^2$, we get the variety of $\bar{B}^{aa'}$ parameterized in terms $\Lambda^a$. If we expand this non-linear expression up to linear order in $\Lambda^a$, we get after some algebra

$$B^{aa'} = [M^a M^{a'} + M^a \Lambda^{a'} + \Lambda^a M^{a'} - (M^a M^{a'}) \oslash (M^+)^2 \odot (M^+ \Lambda^+ + \Lambda^+ M^+)] \oslash (M^+)^2 + O(\Lambda^2) \tag{36}$$

The linear part forms a tangent direction on the $\bar{B}^{aa'}$ variety at $B^{aa'} := M^a M^{a'} \oslash (M^+)^2$.

**Determining the solution dimensions for $J$ and $W$ and $B$.** Now, for each $s$, let $\Gamma^r_{ss'}$ for $r \in \{1 : d_{Js}\}$ span all solutions of $[B^a \odot \Gamma]_{s+} = 0$, which can easily be determined by SVD: $d_{Js}$ is the number zero singular values of matrix $B_{s\cdot}:$, and $\Gamma^r_{s\cdot}$ the corresponding singular vectors. Then, $\bar{J}_{ss'} = J_{ss'} + \sum_r \Gamma^r_{ss'} z_{sr}$ for any $z \in \mathbb{R}^{d_J}$ with $d_J = \sum_s d_{Js}$ is a solution of $[B^a \odot J]_{s+} = \pi(a|s)$.

Similarly, $W^a_{ss'} := M^a_{ss'} + \sum_r \Lambda^{ar}_{ss'} z_{sr}$ with $\Lambda^{ar} := B^a \odot \Gamma^r$ span all solutions consistent with $B^a$ and $\pi$. The solution dimension is $d_W = \sum_s d_{Ws}$, where for each $s$, $d_{Ws}$ is the rank of $\Lambda^{\cdot\cdot}_{s\cdot}$ if interpreted as a $kd \times d_{Js}$ matrix in $as' \times r$. $d_{Ws}$ may be smaller than $d_{Js}$, since unlike $\Gamma^r_{s\cdot}$, $\Lambda^{\cdot\cdot}_{s\cdot}$ may not be full rank.

If we plug $\Lambda^a_{ss'} = \sum_r \Lambda^{ar}_{ss'} z_{sr}$ into (36), after some index manipulation we get

$$\bar{B}^{aa'} = B^{aa'} + \sum_{t=1}^{d} \sum_{r=1}^{d_{Jt}} C^{aa'rt} z_{tr} \oslash (M^+)^2 \oslash (M^+)^2 + O(z^2) \quad \text{with}$$

$$C^{aa'rt}_{ss''} := (M^a_{st} \Lambda^{a'r}_{ts''} + [\Lambda^{ar} M^{a'}]_{ss''} \delta_{ts})[(M^+)^2]_{ss''} - [M^a M^{a'}]_{ss''}(M^+_{st} \Lambda^{+r}_{ts''} + [\Lambda^{+r} M^+]_{ss''} \delta_{ts})$$

$\bar{B}^{aa'}(z)$ is a local parametrization of $B$, and if we drop the $+O(z^2)$, it parameterizes its tangential hyperplane at $B^{aa'}$. Its dimension $d_B$ is the rank of $C$ interpreted as a $k^2 d^2 \times d_J$ matrix in $aa'ss'' \times rt$. Again, $d_B$ may be smaller than $d_W$, since $C$ may not be full rank.

**Remarks.** For $r \in \{1 : d_J\}$, the columns of matrix $C$ span the tangential space of "rescaled" variety $\bar{B}^{aa'}$ at $B^{aa'}$. Again, the columns may not be linearly independent. If $[(M^+)^2]_{ss''} = 0$, then $B^{aa'}_{ss''} = \perp \forall aa'$, hence all such $ss''$ should be ignored in $C^{aa'r}_{ss''}$, but since the corresponding rows in $C$ are 0, they don't contribute to the rank anyway. Numerically, we need to regard all singular values below some threshold as 0. For (to numerical precision) exact $B$, the threshold can be fairly small ($10^{-13}$ in all our experiments). For approximate/learned $B$, the threshold needs to be of the order of the accuracy of $B$.

**Sampling estimate of $d_B$.** A simpler, but less elegant, and more fragile method to estimate $d_B$ is as follows: Fix one solution $J$. Add random noise in direction of the null-space spanned by $\Gamma^r$ so that it stays a solution, i.e. compute $\bar{J} = J + \sum_r \Gamma^r z_{\cdot r}$ for random $z$, and from this, $W$ and $\bar{B}^{aa'}$ for many such random $J$. The resulting point cloud spans covers the solution variety $\bar{B}^{aa'}$. Various tools could be used to analyze this point cloud, e.g. determine its dimension. If $z$ is chosen small, the point cloud concentrates around $B^{aa'}$ and forms a near-linear space, whose dimension $d_B$ can easily be determined by PCA.

**Higher-order $B$ and higher $i$.**    In the same way we can derive the solution dimensions $d_{B\cdots}$ for higher-order $B^{\cdots}$. Also, even though we don't have (yet) an efficient algorithm for solving EqIM($i$) for $i > 1$ if the solution is not unique, we still can determine the dimension of the solutions (at a particular point $M$). Algorithmically already covered is the case of $W$ satisfying EqIM(1)$\land$EqIM(2), whose solution dimension turns out to be $d_W - d_B$. The general procedure is to plug $W = M + \Lambda$ into and linearly expand EqIM($i$) for $i$ we to hold. Together they form a system of linear equations whose solution dimension can be determined by SVD as above.

