**Inverse MDP Models – Supplementary Material**

## A List of Notation

| Symbol | Type | Explanation |
|---|---|---|
| $\perp$ | | undefined |
| $[\![\text{bool}]\!]$ | $\in \{0,1\}$ | =1 if bool=True, =0 if bool=False |
| $d$ | $\in \mathbb{N}$ | number of states |
| $k$ | $\in \mathbb{N}$ | number of actions |
| $i,j$ | $\in \mathbb{N}$ | time index/step |
| $\{i\!:\!j\}$ | $\subset \mathbb{Z}$ | Set of integers from $i$ to $j$ (empty if $j < i$) |
| $s,s',...,s^i$ | $\in \{1\!:\!d\}$ | state at time step 1,2,...,$i$ |
| $a,a',...,a^i$ | $\in \{0\!:\!k\!-\!1\}$ | action at time step 1,2,...,$i$ |
| $b,b',...,b^i$ | $\in \{0\!:\!k\!-\!1\}$ | alternative action at time step 1,2,...,$i$ |
| $a^{:i}$ | $:= aa'...a^i$ | sequence of $i$ actions |
| $a^{<i}$ | $:= aa'...a^{i-1}$ | sequence of $i-1$ actions |
| $\dot{s},\ddot{s}$ | $\in \{1\!:\!\dot{d}\}$ | parts of state, usually $s = (\dot{s},\ddot{s})$ |
| $\varepsilon$ | $>0$ | small number $>0$ |
| $p(...)$ | $\in [0;1]$ | (conditional) probability distribution over states and actions |
| $\pi(a\vert s)$ | $\in [0;1]$ | policy. Probability of action $a$ in state $s$ |
| $M^a,W^a$ | $\in [0;1]^{d\times d}$ | transition-policy tensor $M^a_{ss'} = p(s'\vert sa)\cdot\pi(a\vert s)$ for each action $a$, $W=q$ |
| $B^a$ | $\in [0;1]^{d\times d}$ | inverse 1-step model $B^a_{ss'} = p(a\vert ss')$ for each action $a$ |
| $B^{a++}_{ss'''}$ | $\in [0;1]$ | 3-step first-action inverse model $p(a\vert ss''')$ |
| $J,K,\Delta$ | $\in \mathbb{R}^{d\times d}$ | action-independent $d\times d$ "transition" matrices |
| $^+$ $_+$ | $\cdot^n \to \cdot$ | index summation, e.g. $M^+_{s+} = \sum_{as'} M^a_{ss'}$ |
| $\cdot$ | $(\cdot,\cdot) \to \cdot$ | matrix multiplication: $[AB]_{ss''} = \sum_{s'} A_{ss'} B_{s's''}$ |
| $\odot$ | $(\cdot,\cdot) \to \cdot$ | element-wise multiplication of matrix elements: $[A\odot B]_{ss'} = A_{ss'} B_{ss'}$ |
| $\oslash$ | $(\cdot,\cdot) \to \cdot$ | element-wise division of matrix elements: $[A\oslash B]_{ss'} = A_{ss'}/B_{ss'}$ |
| $\otimes$ | $(\cdot,\cdot) \to \cdot$ | tensor product: $[\dot{M}\otimes\ddot{M}]_{ss'} := \dot{M}_{\dot{s}\dot{s}'}\ddot{M}_{\ddot{s}\ddot{s}'}$ with $s = (\dot{s},\ddot{s})$ and $s' = (\dot{s}',\ddot{s}')$ |

## B Characterizing $M$ and $W$ for which EqIM(1) holds

$$M^a \oslash M^+ = W^a \oslash W^+ \quad \Longleftrightarrow \quad W^a = M^a \odot J \quad \text{with} \quad J := W^+ \oslash M^+$$

That is, $J$ is independent of $a$. Phrased differently

$$\text{For any } M \text{ and } W, \text{ EqIM(1) is satisfied} \quad \textit{iff} \quad W^a \oslash M^a \text{ is independent } a. \tag{14}$$

For a given $M$, this allows to determine all $W$ consistent with EqIM(1), by just multiplying with any $a$-independent $J \geq 0$. Not all $J$ though lead to $W$ consistent with (7). In order to also satisfy (7), $J$ needs to be restricted as follows: With $\Delta_{ss'} := J_{ss'} - 1$, (7) becomes

$$0 \overset{!}{=} W^a_{s+} - M^a_{s+} = \sum_{s'} M^a_{ss'}(\Delta_{ss'}+1) - M^a_{s+} = \sum_{s'} M^a_{ss'}\Delta_{ss'} \tag{15}$$

For each fixed $s$, these are $k$ homogenous linear equations (one for each $a$) in $d$ variables. Given $M$, all and only the $W$ consistent with EqIM(1) *and* (7) can be obtained via $W^a = M^a \odot (1+\Delta)$ with $\Delta$ satisfying $M_{s\cdot}\Delta_{s\cdot} = 0$.

As a special case, $\Delta = 0$ necessarily if and only if the rank of $M_{s\cdot}$ is $\geq d$ for every $s$. This gives the precise conditions as stated in Proposition 1 under which $(i)$ is true. We will next show that EqIM(2) removes this limitation.

 # C  Characterizing $M$ and $W$ for which EqIM(1) and EqIM(2+) hold

From Appendix B we know that the most general Ansatz for $W^a$ satisfying EqIM(1) is $M^a \odot (1+\Delta)$. Plugging this into (31) and expanding in $\Delta$, we get

$$0 = M^a M^+ \odot (M^+)^2 - M^a M^+ \odot (M^+)^2$$
$$+ \ M^a M^+ \odot [M^+(M^+ \odot \Delta) + (M^+ \odot \Delta) \odot M^+] - [(M^a \odot \Delta)M^+ M^a(M^+ \odot \Delta)] \odot (M^+)^2]$$
$$+ \ M^a M^+ \odot (M^+ \odot \Delta)^2 - (M^a \odot \Delta)(M^+ \odot \Delta) \odot (M^+)^2$$

This is a collection of quadratic equations in $\Delta$. The $\Delta$-independent first line is 0. We can write this in canonical form:

$$\Sigma_{kl} A^a_{ss'',kl}\Delta_{kl} = R^a_{kl}(\Delta) \quad \text{with} \tag{16}$$

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

_{s+}^a = \pi(a|s)$, where $\hat{W}^a := B^a \odot \hat{J}$ and $X^a := B^a \odot Y$.

If we plug in $W^a \equiv \hat{W}^a + X^a$ into $B^{aa'}$, we get the variety of $B^{aa'}$ parameterized in terms $X^a$. If we expand this non-linear expression up to linear order in $X^a$, we get after some algebra

$$B^{aa'} = [\hat{W}^a \hat{W}^{a'} + \hat{W}^a X^{a'} + X^a \hat{W}^{a'} - (\hat{W}^a \hat{W}^{a'}) \oslash (\hat{W}^+)^2 \odot (\hat{W}^+ X^+ + X^+ \hat{W}^+)] \oslash (\hat{W}^+)^2 + O(X^2) \tag{36}$$

The linear part forms a tangent direction on the variety at $\hat{B}^{aa'}$.

**Determining the solution dimensions for $J$ and $W$ and $B$.** Now, for each $s$, let $Y_{ss'}^r$ for $r \in \{1 : d_{Js}\}$ span all solutions of $[B^a \odot Y]_{s+} = 0$, which can easily be determined by SVD: $d_{Js}$ is the number zero singular values of matrix $B_{s\cdot}^{\cdot\cdot}$, and $Y_{s\cdot}^r$ the corresponding singular vectors. Then, $J_{ss'} = \hat{J}_{ss'} + \sum_r Y_{ss'}^r z_{sr}$ for any $z \in \mathbb{R}^{d_J}$ with $d_J = \sum_s d_{Js}$ is a solution of $[B^a \odot J]_{s+} = \pi(a|s)$.

Similarly, $W_{ss'}^a := \hat{W}_{ss'}^a + \sum_r X_{ss'}^{ar} z_{sr}$ with $X^{ar} := B^a \odot Y^r$ span all solutions consistent with $B^a$ and $\pi$. The solution dimension is $d_W = \sum_s d_{Ws}$, where for each $s$, $d_{Ws}$ is the rank of $X_{s\cdot}^{\cdot\cdot}$ if interpreted as a $kd \times d_{Js}$ matrix in $as' \times r$. $d_{Ws}$ may be smaller than $d_{Js}$, since unlike $Y_{s\cdot}^r$, $X_{s\cdot}^{\cdot\cdot}$ may not be full rank.

If we plug $X_{ss'}^a = \sum_r X_{ss'}^{ar} z_{sr}$ into (36), after some index manipulation we get

$$B^{aa'} = \hat{B}^{aa'} + \sum_{t=1}^{d} \sum_{r=1}^{d_{Jt}} C^{aa'rt} z_{tr} \oslash (\hat{W}^+)^2 \oslash (\hat{W}^+)^2 + O(z^2) \quad \text{with} \tag{37}$$

$$C_{ss''}^{aa'rt} := (\hat{W}_{st}^a X_{ts''}^{a'r} + [X^{ar} \hat{W}^{a'}]_{ss''} \delta_{ts})[(\hat{W}^+)^2]_{ss''} - [\hat{W}^a \hat{W}^{a'}]_{ss''} (\hat{W}_{st}^+ X_{ts''}^{+r} + [X^{+r} \hat{W}^+]_{ss''} \delta_{ts}) \tag{38}$$

$B^{aa'}(z)$ is a local parametrization of $B$, and if we drop the $O(z^2)$, it parameterize its tangential hyperplane at $\hat{B}^{aa'}$. Its dimension $d_B$ is the rank of $C$ interpreted as a $k^2 d^2 \times d_J$ matrix in $aa'ss'' \times rt$. Again, $d_B$ may be smaller than $d_W$, since $C$ may not be full rank. for $r \in \{1 : d_J\}$ spans the tangential space of rescaled variety $B^{aa'}$ at $\hat{B}^{aa'}$. Again, they may not be linearly independent, If $[(W^+)^2]_{ss''} = 0$, then $B_{ss''}^{aa'} = \perp \forall aa'$, hence all such $ss''$ should be ignored in $C_{ss''}^{aa'r}$, but since the corresponding rows in $C$ are 0, they don't contribute to the rank anyway.

**Sampling estimate of $d_B$.** A simpler, but less elegant, and more fragile method to estimate $d_B$ is as follows: Fix one solution $\hat{J}$. Add random noise in direction of the null-space spanned by $Y^r$ so that it stays a solution, i.e. compute $J = \hat{J} + \sum_r Y^r z_{\cdot r}$ for random $z$, and from this, $W$ and $B^{aa'}$ for many such random $J$. The resulting point cloud spans covers the solution variety $B^{aa'}$. Various tools could be used to analyze this point cloud, e.g. determine its dimension. If $z$ is chosen small, the point cloud concentrates around $\hat{B}^{aa'}$ and forms a near-linear space, whose dimension $d_B$ can easily be determined by PCA.

**Higher-order $B$ and higher $i$.** In the same way we can derive the solution dimensions $d_{B\cdots}$ for higher-order $B^{\cdots}$. Also, even though we don't have (yet) an efficient algorithm for solving EqIM($i$) for $i > 1$ if the solution is not unique, we still can determine the dimension of the solutions (at a particular point $\hat{W}$). Algorithmically already covered is the case of $W$ satisfying EqIM(1)$\wedge$EqIM(2), whose solution dimension turns out to be $d_W - d_B$. The general procedure is to plug $W = \hat{W} + X$ into and linearly expand EqIM($i$) for $i$ we to hold. Together they form a system of linear equations whose solution dimension can be determined by SVD as above.

# Q   Counter-Examples in Related Work

In Section 3 we presented a counter-example to questions (i,iii,v). Question (i) (i.e. Can $M$ be inferred from $B^a := M^a \oslash M^+$?) has been implicitly addressed in previous work. In [EMK$^+$22, App.A.3] the authors present a counter-example to the claim that a state representation constructed

via an inverse model (i.e. two states have the same representation iff they yield the same inverse distribution for all of their possible successor states) is sufficient for representing a set of policies that differentially visit all states[1]. This fails whenever two states are aliased by the inverse model.

Note that this failure of state representation learning implies a negative answer to our question (i), as $W$ would differ from $M$ on these aliased states. Unlike our counter-example, theirs involves deterministic forward dynamics, and therefor buttresses our claims by showing that $M$ cannot always be inferred even in this simpler case. Similar to our counter-example in Section 3, [MHKL20] proposes a stochastic counter-example to inverse modeling for state representation learning.

In general, the transferability of these counter-examples suggests a strong relationship between literature on using single-step inverse models for state representation learning and using them for inferring the forward model. It is an interesting open question whether or not algorithms for representation learning on the basis on multi-step inverse models (like those put forward in [EMK$^+$22]) might be used to shed light on the questions put forward here and vice versa.

## R  Relevance for Planning

In Section 1, various streams of applied work were highlighted; here we focus on spelling out the overarching impact that compositional inverse models (an affirmative answer to question (iv)) would have for planning problem.

Many forms of planning involve the evaluation of candidate $i$-step action sequences (e.g. model predictive path integral control [WDG$^+$16]). Ideally, all possible action sequences would be evaluated, but as the space of $i$-step action sequences grows exponentially in $i$, this is often intractable.

Access to the $i$-step inverse distribution $p(a...a^i|s...s^{i+1})$ allows determining the subset of action sequences that likely reach state $s^{i+1}$ post-execution (e.g. those whose probability is above some threshold). It is often the case that only action sequences that are distinguishable in this way are of interest (e.g. goal-reach tasks), thus access to an inverse model of the appropriate horizon allows for filtering candidates. This filtering method is a particularly appealing approach when the cost/reward function is initially unknown and frequently changes.

While this idea has already seen scalable implementations [MJR15], these rely on short, fixed horizons since they directly learn all inverse models of step-size up to the horizon, which is data-inefficient for large horizons. If inverse models could be composed, then longer, variable horizons could be used while only learning a short horizon inverse model by inferring the longer horizon models as needed. Our work shows that this is possible, but with exceptions and a more practical composition algorithm being outstanding.

---

[1]Technically, as per their Definition 2, this 'policy cover' need only account for all 'endogenous' states. But omit the 'exogenous' states from their counter-example and it can be seen to address our question (i).