# OpenReview forum: "Uniqueness and Complexity of Inverse MDP Models"
_NeurIPS.cc/2022/Conference — NeurIPS 2022 Submitted_

### Official Review · Reviewer_SKHG · 2022-06-30

**Rating:** 3
**Confidence:** 2
**Soundness:** 2 fair
**Presentation:** 1 poor
**Contribution:** 2 fair

**Summary:**

This paper analyzes the prospects of inferring meaningful inverse models of discrete-time controlled Markov processes. In particular, the paper focuses on determining the existence/uniqueness properties of these inverse models, and investigate the computational complexity of computing them.

**Questions:**

**General Question**: As the authors acknowledge, the setting here is not that of inverse MDPs, but inverse controlled Markov processes. I'd argue actually that it's simply inverse Markov chains, since the policy appears to be fixed everywhere. How do the studies in this paper differ from those in the inverse kinematics literature?

  **Motivation: causal inference, line 24**: It is written that "An
  inverse model captures the likelihood that an action $a$ was *the
  cause* of the transition...", but is this truly correct? To my
  understanding, such a conditional model is incapable of inferring
  causation. Wouldn't one need some sort of counterfactual analysis as
  well?

  **Marginalizing variables in transition tensors**: On lines 120 and 121,
  I believe there is an inconsistency in the notation. In the Notation
  section, it is writen that $M^a_{\cdot\cdot}$ is the matrix at
  dimension $a$, but here it is written $M^a$. I'm assuming $M^a,
  M^a_{\cdot\cdot}$ represent the same thing, but then maybe the
  $M^a_{\cdot\cdot}$ notation should be avoided altogether.

  **Challenges with the backward model**: Line 127 says $B^{a+}\neq B^a$,
  can this be expanded upon? This is unintuitive, at least to me. After
  line 121, it says $B^a_{ss'} = p(a\mid ss')$, and after line 129 it
  says $B^{a+}_{ss''} = p(a\mid ss'')$, thus

  \begin{equation*}
  \begin{aligned}
  B_{ss'}^{a+} = p(a\mid ss')  =B_{ss'}^a
  \end{aligned}
  \end{equation*}

  Which implies that $B^a = B^{a+}$ since all of their entries are
  equal.

  **The Questions**: The six questions, first stated on line 46, are very
  confusing. Firstly, they're stated in different terms several times,
  which is confusing on its own. Why not list the questions only once,
  in the most intuitive way? Moreover, the questions are too complicated
  individually. I think it would be much clearer if you don't state all
  of these questions explicitly, and rather prove the answers as
  theorems/lemmas/propositions. It's very hard to remember which
  question is which as you go through the text, and it's actually very
  hard to digest what the paper is trying to do with such a dense list
  of questions. In my opinion, the motivation for the paper is clear
  enough on its own, and I don't think it'll be difficult to convince
  most readers that understanding these inverse models is an important
  task. But when the exact meaning of "understanding" is stated so
  verbosely, it actually detracts from the clarity of the paper. One
  example that is particularly confusing appears on line 139, where it
  says "Questions (i) - (vi) also have multiple variations", followed by
  a list of three questions. How do you go from 6 questions to 3, which
  correspond to which? Now you're actually assuming that the reader will
  recall at least 9 questions.

  Moreover, if you must include these questions, it would be
  tremendously more convenient if they can be hyperlinked.

  **Line 149**: Another rephrasing of the questions here says

  > One way to rephrase the questions is whether $f(M) = f(W)$
  > implies $M=W$ or $g(M) = g(W)$ for all (or most or some)
  > $M$ and $W$.

  I don't understand this question at all. The property that $f(M) =
  f(W)\implies M = W$ is more a property of the function $f$ than the
  objects $M,W$. Indeed, this property is known as
  "injectivity". Likewise, "whether $f(M) = f(W)$ implies $g(M) = g(W)$"
  is also highly dependent on the particular properties of
  $f,g$. Currently $f,g$ are only assumed to be arbitrary functions, so
  it is really unclear to me what the purpose of this question is. I had
  a clearer understanding of the questions before reading this.

  **Equations (8) and (9)**: Surely the notation can be improved here, it
  is extremely difficult to parse. Why not use product notation
  ($\prod$)? Also, in equation (9), what is $B^{a+\dots+}$? Are there
  $i$ dots? Back to my previous comment, I believe $B^{a+\dots+}\equiv
  B^a$ unless there's an inconsistency in the notation.

  Later, on line 161, the term $\text{EqIM(1)}$ is mentioned, but where
  is that defined? Only equations $\text{EqIM}(ia)$ and
  $\text{EqIM}(i+)$ are defined.

  **Example violating (i, iii, v)**: It appears that what you're showing
  is that for random variables $X, Y$ distributed by $\mu_X, \mu_Y$,
  there may exist several distinct random variables $(X, Y)$ with
  marginals $\mu_X, \mu_Y$. This is well known.

  **Line 169**: I don't understand how this relates to (i), (iii), (v) at
  all. You're assuming that two MDPs encoded by $M,W$ have the same
  transition dynamics, and then based on that, the ability to infer
  multistep models from fewer-step models is guaranteed? Something is
  missing here. How does $W$ fit in?

  **Line 171**: It says "Note that $M_{ss'}^a\equiv p(s'\mid sa)$
  independent of $a$ implies $M_{s+}^a$ independent of $a$..." -- this
  is only true if it holds /for every pair/ $(s, s')$. Also, this whole
  "degenerate case" is fairly trivial: $M$ and $W$ are only related via
  $\pi(a\mid\cdot)$, so if they're both actually independent of $a$,
  then of course any equality of this sort must imply that the MDPs are
  identical.

  **Line 175**: What does it mean for tensors to be "nearly independent"
  of $a$?

  **Line 178**: Similarly to the "independent of $a$" case, why is this
  even an interesting case to study? As pointed out, the consequences
  are fairly obvious.

  Altogether, I didn't find section 3 to be very informative.

  **Non-linear manifold**: On line 212, it says that $B^{aa'}$ forms a
  non-linear manifold of dimension at least $d(d-k)$. But what is the
  "dimension" of a non-linear manifold? Perhaps there is a definition
  for this, but I doubt it's widely known to the Neurips audience. I'd
  much prefer to have seen this definition in the background section in
  place of the algebraic properties of tensors.

  **Line 214**: I believe this logic is flawed. It says "... this now
  gives an over-determined system which generally has no solution. But
  by assumption, $M$ is a solution, which gives hope that there may be
  only one or a finite number of solutions". What is being said here is
  that the situation only makes sense if the "over-determinacy" is
  redundant, so that it's not actually over-determined. This doesn't
  give any hint about the number of solutions. There may very well be
  infinitely-many solutions regardless of the fact that the system is
  over-determined.

  **Figure 1**: This figure is never mentioned in the text, it would be
  nice to have some discussion of the results it's presenting.


Minor issues
============

  **Contents section**: On line 88 "Appendix A" is mentioned, but then
  subsequent appendices are written "(B)", for instance, which is a
  little confusing.

  **Matrix notation**: On line 105, it is written "Capital letters... are
  used for $d\times d$ matrices over $[0;1]\subset\mathbf{R}$...".  Is
  $[0;1]$ the closed unit interval? Why not write "Capital
  letters... are used for matrices in $[0, 1]^{d\times d}$"?

  **Tensor notation**: On line 109,

  > Let $\odot$ denote element-wise (Hadamard)
  > multiplication... and similarly $\oslash$, while (no)
  > $\cdot$ represents (conventional) matrix multiplication
  > and has operator preference over $\odot$ and $\oslash$.

  I don't understand this at all. What is $\oslash$? What is $\cdot$?
  I'm assuming that by "similarly $\oslash$" it is meant that $\oslash$
  is element-wise division (this should be stated, because the sentence
  doesn't make sense otherwise), and that $\cdot$ represents matrix
  multiplication. But what does "while (no) $\cdot$" mean?

  **Algebraic properties of tensors**: Does this really need to be
  included here? It's hard to read math packed into a paragraph like
  this. Also many of these properties are very intuitive (i.e., I think
  just about every reader will have inuition for the $D\odot\text{Id}$
  property without it being stated), and then some properties are highly
  abstract (matrices form a ring...). I think it's fair to assume that
  readers of this paper understand how to do algebra with matrices and
  tensors, even if they don't know what a ring is.


**Limitations:**

The authors have addressed the limitations of their methods.

**Strengths And Weaknesses:**

Unfortunately, several technical parts of this paper concern me. There are several mathematical statements that I believe to be invalid (see the Questions section). Moreover, I found the structure of the paper to be extremely confusing, especially the recurring "questions" which are rephrased several times, some of which (as far as I'm concerned) are not rephrased correctly. Finally, the notation used in the paper is very difficult to parse.

---

> ### Author Response · Authors · 2022-08-06
> **Response to Reviewer SKHG**
>
> Thank you for your time and detailed comments. We have addressed several of your primary concerns below, and the overarching concern of clarity in the new draft of the paper. We hope you'll see that there aren't any known mathematical issues once your notational confusion is resolved and raise your score accordingly. Additionally, the new draft bolsters the empirical side of the work, which we hope you'll appreciate.
>
> > General Question: As the authors acknowledge, the setting here is not that of inverse MDPs, but inverse controlled Markov processes. I'd argue actually that it's simply inverse Markov chains, since the policy appears to be fixed everywhere. How do the studies in this paper differ from those in the inverse kinematics literature?
>
> We admit ourselves that [inverse] controlled Markov process would be a more apt name, but even with our fixed policy our setup is NOT a "[inverse] Markov chain". Markov chains are only about state sequences. There are no actions. We infer action probabilities from states sequences. Most of our results do not rely on the assumption of fixed policies. In particular, our most significant results are counter-examples which remain counter-examples in this broader class of problems (in brief, if you can't infer a property of how a fixed policy interacts with the environment, then you can't infer it if the policy is also changing).
>
>
> > Marginalizing variables in transition tensors: On lines 120 and 121, I believe there is an inconsistency in the notation...
>
> The dots in the M^{⋅}_{⋅⋅} notation indicate arguments. Hence  is a 3-tensor whereas M^{a}_{⋅⋅} is a matrix and M^{a}_{s s'} is a scalar. This is used in many places, such as Equation 8.
>
> >Challenges with the backward model: Line 127 says , can this be expanded upon? This is unintuitive, at least to me. After line 121, it says
> , and after line 129 it says
> , thus
>
> This is incorrect. p(a|ss") is the probability of reaching s" from s with a sequence of 2 actions, first one being a and second one being any. This is precisely also the difference between B^a_{ss'}=p(a|ss') and B^{a+}_{ss"}=p(a|ss")=∑_a' p(aa'|ss").
>
> > Line 149: Another rephrasing of the questions here says
> One way to rephrase the questions is whether  implies  or  for all (or most or some)  and .
> ... Indeed, this property is known as "injectivity". Likewise, "whether  implies " is also highly dependent on the particular properties of . Currently  are only assumed to be arbitrary functions, so it is really unclear to me what the purpose of this question is.
>
> We are not talking about arbitrary functions here. Rather, we are noting that the various sorts of inverse models in questions iii-vi) are all functions of M. Thus, these questions concern whether this implication holds *for the specific functions we're considering*.
>
> > Equations (8) and (9): Surely the notation can be improved here, it is extremely difficult to parse. Why not use product notation ()? Also, in equation (9), what is ? Are there  dots? I believe  unless there's an inconsistency in the notation.
>
> The fact that these are different matrices denoted by their superscript makes product notation similarly unruly. The ellipsis in between elements in the superscript denote arbitrary additional elements. That is to say, we're talking about multi-step models of arbitrary length. As per our previous response, you were mistaken about the equality in question.
>
>
> > "Questions (i) - (vi) also have multiple variations" "How do you go from 6 questions to 3".
>
> No, each question has 3 (now 4) variations, i.e. it is 6 times 3, not 6 plus 3 questions.
>
> > Line 169: I don't understand how this relates to (i), (iii), (v) at all. You're assuming that two MDPs encoded by  have the same transition dynamics, and then based on that, the ability to infer multistep models from fewer-step models is guaranteed? Something is missing here. How does  fit in?
>
> This is just noting one of the conditions under which all inferences hold. We then go on to consider cases where M != W. We agree this is rather degenerate (hence the section title), but included for completeness. This has been moved to the appendix.
>
> > Line 171: It says "Note that
>  independent of  implies
>  independent of ..." -- this is only true if it holds /for every pair/ . Also, this whole "degenerate case" is fairly trivial:  and  are only related via , so if they're both actually independent of , then of course any equality of this sort must imply that the MDPs are identical.
> Line 175: What does it mean for tensors to be "nearly independent" of ?
> Line 178: Similarly to the "independent of " case, why is this even an interesting case to study? As pointed out, the consequences are fairly obvious.
>
> We agree these cases are fairly obvious and have moved them into the appendix, though we believe that for a novel setup it is good practice to establish simple facts first to establish some common ground and basic understanding before diving into intricate matters.

---

> > ### Comment · Reviewer_SKHG · 2022-08-08
> > **Revision?**
> >
> > Thanks to the authors for the in-depth response. Some of my concerns were addressed, however I am still confused by several things. In particular, I still don't understand the correspondence between all of the (18+, as you point out) questions. I understand that, with regard to the property that I suggested was simply injectivity, that you're restricting the functions $f, g$ to specific functions -- however, I am still not convinced of the relevance of the result. I also still don't understand the relationship between $M$ and $W$, nor the meaning of the dimension of the nonlinear manifold and the conclusion you draw from that. I was hoping to see a revision that attempts to clear up some of the confusing parts, but I see no revision here.

---

> > > ### Author Response · Authors · 2022-08-08
> > > **Revision already uploaded**
> > >
> > > The revision was already uploaded, and I just confirmed it worked by pressing the "pdf" button at the top of the page. It should be easy to tell because it's merged the main and appendix so the total pdf length is 28 pages. Please let me know if you are still unable to access it.

---

> > > > ### Comment · Reviewer_SKHG · 2022-08-09
> > > > **Revision**
> > > >
> > > > My mistake, I was checking the "revisions" tab this whole time! I am looking through the revision at the moment.

---

> > > > ### Comment · Reviewer_SKHG · 2022-08-09
> > > > **Post-Revision Updates**
> > > >
> > > > Upon reading the revision, some questions have been clarified, but unfortunately I still find the text very unclear. At the very least, I strongly believe the notation should be overhauled and the recurring questions should be consolidated, as I genuinely find the text to be extremely difficult to parse. That said, the authors' responses (to my comments and those of the other reviewers) has helped clear up some of my confusions, and I have slightly increased my score as a result.
> > > >
> > > > I still believe the text suffers from the following main issues (in addition to the notation / organization):
> > > >
> > > > 1. Many of the results are fairly predictable / evident. It is not surprising that in many (most?) cases, inverse models cannot be recovered. As I see it, the only condition for which the paper asserts that an inverse model can be found (rank >= d) is a fairly simple fact from linear algebra.
> > > > 1. Some of the logical arguments still seem askew. Particularly, lots of the dimensional analysis arguments are imprecise and/or speculative (particularly, I still find those starting on line 202 of the revision to be very hand-wavy). Lots of these arguments include phrases like "gives hope" and "the solution may be unique", and introduce ad-hoc assumptions, which makes it difficult to assess what, if any, conclusion actually follows from the premises.
> > > > 1. Many of the questions are imprecise / confusing, and it's never clear to me what the answers are. In the revised paper, it says "One way to rephrase the questions is whether $f(M) = f(W)$ or $g(M) = g(W)$ for all (or most or some) $M$ and $W$". How do you quantify "most or some", and how does that fit in to the original question? Doesn't the quantifier (all, most, some) drastically change the implications of the question? Where are each of these questions answered? Moreover, while I think I understand what these $f,g$ functions are supposed to represent now after seeing the authors' response, it should be stated much more explicitly.
> > > > 1. It is never clear to me what the answers to the original questions are. Perhaps this is because each question is broken down into a bunch of sub-questions each having their own assumptions downstream, and the answers are never consolidated or clearly linked back to the original questions.
> > > > 1. Related to the previous point, it's not clear to me what the takeaways of this paper are supposed to be.
> > > >
> > > > Altogether, I do think inverse models can be quite useful and are interesting objects to study. Ultimately, however, I am struggling to assess the significance of many of the results in this paper.

---

> > > > > ### Author Response · Authors · 2022-08-09
> > > > > **Obviousness**
> > > > >
> > > > > > Many of the results are fairly predictable / evident. It is not surprising that in many (most?) cases, inverse models cannot be recovered.
> > > > >
> > > > > Is it really obvious that, given a k-step inverse model, you can't (in general) recover the (k+1)-step inverse model? We proved this in Appendix K, and would appreciate seeing a citation or simpler construction in the case that we were mistaken and this was already common knowledge.
> > > > >
> > > > > Similarly, is our algorithm for approximately recovering M via linear relaxation (Section 5) seems novel to us, so we'd appreciate a more detailed explanation by which it appears "predictable / evident".

---

> > > ### Author Response · Authors · 2022-08-08
> > > **Clarification**
> > >
> > > >  I am still not convinced of the relevance of the result.
> > >
> > > Appendix B in the revision is dedicated to illustrating how, specifically, answering these questions is relevant to planning.
> > >
> > > > I also still don't understand the relationship between  M and W
> > >
> > > M and W are two different MCP forward dynamics, and they are used extensively throughout the paper since the majority of questions concern whether or not these two MCPs share some property based off of same other shared property (e.g. does M=W if the 1-step inverse models match?). It's often helpful to think of W as a learned model whereas M is the ground truth environmental dynamics. Under this framing, the questions concern whether or not certain partial models are sufficient for some purposes (e.g. if I learnt a perfect 1-step inverse model, can I use it to recover the forward dynamics?).
> > >
> > > >the meaning of the dimension of the nonlinear manifold and the conclusion you draw from that.
> > >
> > > From the results section of the revision:
> > > "When the solution to an inverse model (B2) given only B1 is not unique, we can characterize the solution space in terms of its manifold dimension. By comparing this to the dimension of that of the inferred forward model (W), we can see that our algorithm has narrowed down the space of inverse models further."

---

### Official Review · Reviewer_mjPn · 2022-07-10

**Rating:** 6
**Confidence:** 2
**Soundness:** 3 good
**Presentation:** 2 fair
**Contribution:** 3 good

**Summary:**

The authors pose the question of whether or not the 1-step inverse model, along with the policy \pi, determine the multi-step inverse model p(a | ss^i) - or if the forward dynamics of an environment can be inferred by the inverse model plus policy. Besides this question, the work also looks to analyze other cases of whether or not the 1-step inverse model and policy can determine other forms of inverse dynamics, such as the multi-step inverse models, and present analysis and empirical evidence for when these cases hold or when they do not hold.

**Questions:**

As an aside, many of these questions will be clarification questions - hopefully these questions will help polish the paper so that it’s more clear for a more broad set of readers.

Section 2:
For the end of section 2, could you give an example of how one of the 6 questions can be inferred from f(M) = f(W) -> M = W? I find it a bit hard to follow what this other MDP W relates to B^a and M^a.

Your overloaded notation for EqIM is a bit confusing. You defined EqIM(ia) and EqIM(i+), but not EqIM(i).

Section 4:
For the introduction to this section, it would be nice to relate back to the 6 questions the authors were trying to answer and how EqIM(1) and EqIM(2) relates to those questions. This also relates back to my question in Section 2 about your definition of EqIM and how it is another way of posing the questions.

Section 5:
Again, it would be good to relate back to the questions. Correct me if I’m wrong but this section tries to answer (iii) to (vi)?

Section 6:
I don’t see the experimental details listed anywhere - how do you generate these randomized grid worlds?

Also instead of just showing samples of the true vs inferred forward dynamics matrices, I think it would be more prudent to show statistics gathered from experiments, instead of just one or two examples. See limitation for more suggestions for experimental section.


**Limitations:**

One limitation of this approach is the scalability to large-scale environments. While this is a limitation, I don’t believe this is any issue in the analysis provided in this work, or with the claims made. That being said, it would be very interesting to see how well some of these theoretical claims hold in larger environments, with more complicated dynamics/inverse dynamics that can’t be feasibly represented with tensors.

Another limitation of this work is with the experiments and empirical results. While the do run experiments to back up the theory they develop, they don’t provide details of their experimental set up, nor do they present any meaningful statistics with regards to their main result - showing one or two examples from generated grid worlds in the main work (the first grid world does not match the canonical 4-room by the way - it is something akin to it), and presenting a few plots over noise added for noisy inverse models. There is also stochasticity from the grid world generation - was there a lot of variation in terms of inferred dynamics models in the deterministic case? How were the grid worlds generated? It seems that (based on the second grid world presented) the dynamics of the grid world would be quite sparse. As mentioned earlier, presenting a more detailed account and analysis of the experiments done would improve the paper.


**Strengths And Weaknesses:**

I believe that the authors make a meaningful and well-justified contribution towards understanding the theory behind inverse dynamics models and reinforcement learning as a whole. The authors present interesting theoretical insights into the realizability and limitations of inverse models, along with algorithms to potentially calculate both the two-step inverse model and forward dynamics model from the inverse model plus policy.

In terms of weaknesses, one weakness that comes to mind from the perspective of someone not too familiar with this form of reinforcement learning theory is the writing itself - the writing style is hard to follow in many parts, taking multiple passes for even some of the non-notation heavy parts - this seems to come from the somewhat informal tone the authors take in this manuscript, that sometimes adds a bit of cognitive burden for a reader of this paper. While the authors might be very familiar with the notation and work, I believe it would improve the polish of the paper quite a bit if the authors were a bit more careful and precise with their use of language and notation.

Finally, I believe the experimental section has a decent amount to be desired. While I don’t think there’s anything wrong with the fundamental methodology of their experiments, the authors have left out quite a bit of experimental details and forego presenting actual statistics of experimental results - instead they simply present two examples in the main work. While they do show a plot detailing the effect of adding noise to the inverse dynamics models, my biggest question is with the grid world generation itself. It seems that most of the grid worlds generated would have very sparse dynamics (like in the second grid world presented) - many of the positions/states would be unreachable from many other states. It would be more clear if the authors presented more details on their experiments done to validate their theory.

Overall, while I not intimately familiar with the theory presented in this paper, I do think this work provides a valuable contribution to the field in terms of understanding the limitations of inverse dynamics models.

---

> ### Author Response · Authors · 2022-08-06
> **Response to Reviewer mjPn**
>
> Thank you for your time and consideration. We largely agree with your assessment and hope the latest draft reflect a marked improvement. In particular, the experimental section has been expanded to include new results and more discussion, with an entire experimental details appendix (P) added as well. We've addressed your specific points below, but believe you'll be more persuaded by checking out the updated draft.
>
> >Section 2: For the end of section 2, could you give an example of how one of the 6 questions can be inferred from f(M) = f(W) -> M = W? I find it a bit hard to follow what this other MDP W relates to B^a and M^a.
>
> Good question! This is basically just to note that inverse quantities like B^a are functions of the forward dynamics M (or W). So, we're effectively asking if, for this very specific set of functions, whether this implication holds. For example, question i) f would be the function transforming the forward dynamics into the inverse dynamics. If we apply this function to two forward dynamics (M and W) and the resulting distributions are the same, then does that imply the forward dynamics the same? (The answer is no in general). This framing makes the notion of "can X be inferred from Y" precise in a way that exploits knowledge of our problem (Y is a function of X).
>
> > Also instead of just showing samples of the true vs inferred forward dynamics matrices, I think it would be more prudent to show statistics gathered from experiments, instead of just one or two examples. See limitation for more suggestions for experimental section.
>
> We agree! This is now FIgure 2.
>
> >How were the grid worlds generated? It seems that (based on the second grid world presented) the dynamics of the grid world would be quite sparse
>
> Indeed, they were actually generated with constant sparsity in some sense. We simply fixed the number of "wall locations" and sampled their location on a square grid uniformly at random. We did consider varying this up a bit, but initial attempts at other generation procedures didn't yield interestingly different results, and these environments are too 'toy' to be cared about for their own sake. We do believe scaling to more interesting domain is important future work, but would require non-trivial extensions/approximations to make tractable.
>
> Thank you again for your review. While your current score is already over the acceptance threshold, we'd still very appreciate it if you increased your score based off of these responses and the considerable effort that went into the latest draft.

---

> > ### Comment · Reviewer_mjPn · 2022-08-09
> > **Re: Response**
> >
> > Thank you for the response and the clarifications!
> >
> > While this response his helpful for my understanding of your work, I will be maintaining my score of a weak accept - this score also reflects the certainty in my assessment of this work - I'm not intimately familiar with the background behind this subarea of reinforcement learning theory, and don't feel confident in giving a better score.

---

### Official Review · Reviewer_5AUZ · 2022-07-11

**Rating:** 3
**Confidence:** 4
**Soundness:** 3 good
**Presentation:** 2 fair
**Contribution:** 2 fair

**Summary:**

This paper concerns the question of if and when the transition dynamics of an MDP can be determined based on a policy together with its corresponding inverse dynamics. In the case of single-step inverse dynamics, they find that this is not the case, unless the number of actions is greater than the number of states, and each action corresponds to a full-rank transition matrix. The paper also studies a number of similar questions.

**Questions:**

What is the main (practical or theoretical) significance of these results? Which result is the most important?

**Limitations:**

The limitations are discussed, though this discussion is dispersed throughout the paper, rather than being collected in one place. There is no discussion of potential negative societal impacts (but I also do not think that is needed in this case).

**Strengths And Weaknesses:**

Originality:

This question has, to be best of my knowledge, not been examined previously (at least not as thoroughly as in this paper). However, some of the results seem to be reasonably straightforward (for example, the fact that the transition dynamics cannot necessarily be inferred from a given policy together with its corresponding one-step inverse model is immediately clear if one considers any policy that samples actions independently from the state).


Quality:

I have not checked most of the calculations carefully, but based on what I have checked, the maths appears to be sound.


Clarity:

The clarity of the writing is reasonable, but could be improved. For example, the text takes time to explain elementary notions from linear algebra, and other facts that I think the reader could reasonably be assumed to be familiar with (such as e.g. the fact that matrices form a ring under ordinary matrix addition and matrix multiplication, etc). I also think the paper could be better organised. For example, the introduction does not make it clear what are the main results of the paper, or what their significance is, and it takes some digging to get a full overview of things.


Significance:

I think that the significance of the results is not well explained in the paper, and this is one of its main weak points, in my opinion. The introduction states that the presented results are relevant to causal inference, but the example that is given afterwards is just an instance of normal retrodiction, which doesn't engage with most of the issues discussed in the current literature on causal inference. It is also stated that these results could be of interest for automatic planning, but this seems unclear to me (at least in their current form). It is also stated that the inverse model could be much smaller than the forward model, but it is again not clear to me why that is (practically or theoretically) significant.


Summary:

I have not recommended this paper for publication. The main reason for this is that the problems which are studied here are fairly niche, and it is somewhat unclear what their significance is. Moreover, many of the problems which are stated in the paper are not solved, and some of the ones which are solved are somewhat straightforward. For this reason, I do unfortunately not think that these results will be of significant interest to the wider NeurIPS readership.

---

> ### Author Response · Authors · 2022-08-06
> **Response to reviewer 5AUZ**
>
> Thank you for your time. We've addressed your specific concerns below, and we hope the overarching issues of clarity and significance are addressed by the latest revision. The new version of the paper improves clarity by moving more straightforward elements to the appendices and focusing in on the core contributions. The motivation section has been expanded to include more details on the relationship to planning and other potential application areas. And most importantly, additional experimental results have been included which we hope cement the algorithms introduced as being significant contributions (in addition to the theoretical results which primarily show the impossibility of some inferences).
>
> > I have not recommended this paper for publication. The main reason for this is that the problems which are studied here are fairly niche, and it is somewhat unclear what their significance is. Moreover, many of the problems which are stated in the paper are not solved, and some of the ones which are solved are somewhat straightforward. For this reason, I do unfortunately not think that these results will be of significant interest to the wider NeurIPS readership.
>
> We want to emphasise that several of the counter-examples introduced here represent fundamental limits on the ability to translate knowledge of short-term inverse dynamics into long-term inverse dynamics. They are proofs of impossibility, not merely incomplete attempts at a solution. We agree that the initial presentation mixed together trivial results (included for completeness and didactic purposes) with complex and novel results, and we hope we've improved the presentation in the latest version.
>
> > I have not checked most of the calculations carefully, but based on what I have checked, the maths appears to be sound.
>
> We believe that further inspection might highlight the significance of these results. While we acknowledge that the proofs are quite long and complex, we would highly suggest going through Appendix K (in the new draft), as we believe it is representative, and hope you'll agree that it is 1) correct, 2) novel, and 3) answers a reasonable question.
>
> If so, we hope you'll reconsider your score. In particular, note that it's generally quite hard to predict a paper's significance in advance, so  we urge you to judge this work on the based largely on whether or not it accomplishes what it sets out to do.

---

### Meta-Review · Area_Chair_VnyZ · 2022-08-26

**Recommendation:** Reject
**Confidence:** Less certain

**Metareview:**

In this paper, the authors concern the question of whether or not the 1-step inverse model, along with the policy, can determine the multi-step inverse model - or if the forward dynamics of an environment can be inferred by the inverse model along with the policy. The authors also dwell into other related questions. The authors provide insightful answers and discussions to these questions, but the paper is unfortunately written in a very unconventional format which makes it very hard to read. I hope the authors find the reviewer comments below help restructuring the paper to better enlighten the readers.

**Award:**

No

---

### Decision · Program_Chairs · 2022-09-14

Reject